# Practical intelligent diagnostic algorithm for wearable 12-lead ECG via self-supervised learning on large-scale dataset

Jiewei Lai [1,2,3], Huixin Tan[1,2,3], Jinliang Wang[4], Lei Ji[5], Jun Guo[6], Baoshi Han[6], Yajun Shi[6], Qianjin Feng [1,2,3] ✉ & Wei Yang [1,2,3] ✉

Cardiovascular disease is a major global public health problem, and intelligent diagnostic approaches play an increasingly important role in the analysis of electrocardiograms (ECGs). Convenient wearable ECG devices enable the detection of transient arrhythmias and improve patient health by making it possible to seek intervention during continuous monitoring. We collected 658,486 wearable 12-lead ECGs, among which 164,538 were annotated, and the remaining 493,948 were without diagnostic. We present four data augmentation operations and a self-supervised learning classification framework that can recognize 60 ECG diagnostic terms. Our model achieves an average area under the receiver-operating characteristic curve (AUROC) and average F1 score on the offline test of 0.975 and 0.575. The average sensitivity, specificity and F1-score during the 2-month online test are 0.736, 0.954 and 0.468, respectively. This approach offers real-time intelligent diagnosis, and detects abnormal segments in long-term ECG monitoring in the clinical setting for further diagnosis by cardiologists.

The electrocardiogram (ECG) is a noninvasive method that uses an electrocardiograph to record changes in electrical activity of the heart during each cardiac cycle, detected from the skin surface. It reflects the physiological state of different parts of the heart from multiple perspectives, and it is the most important means for cardiologists to diagnose abnormal cardiac rhythms. According to statistics[1], deaths from cardiovascular diseases (17.9 million) overwhelmingly surpass those from cancer (9.3 million), and cardiovascular disease seriously threatens the lives of countless people around the world. The middle-to-old-age population is at high risk of cardiovascular disease, and the increased population aging in modern society means that the medical burden is gradually increasing. The early detection and medical intervention of arrhythmia are of great significance to a patient's excellent prognosis. It is easier to detect transient and valuable arrhythmia fragments for the first time using long-term continuously

wearable ECG monitors, which can improve patients' health status and prevent more serious complications. The large number of users and prolonged monitoring periods have greatly increased the burden for cardiologists to make diagnoses, aggravating the shortage of medical resources and making the demand for intelligent ECG diagnosis systems more urgent.

The standard 12-lead ECG used in hospitals is the Wilson lead system, which usually takes <1 min to collect, with the patient in a resting state. Wearable ECGs for home monitoring use the Mason-Likar lead system, which requires users to wear electrodes or sensors independently (Fig. 1a). Wearable ECGs are far more prone to artifacts than standard ECGs due to daily human activity; these artifacts include myoelectric artifacts, motion artifacts and electrode shedding. As shown in Fig. 1d, the signals from high-quality wearable ECGs are clear and smooth. In Fig. 1f, the myoelectric artifact appears as irregular

[1]School of Biomedical Engineering, Southern Medical University, Guangzhou, China. [2]Guangdong Provincial Key Laboratory of Medical Image Processing, Guangzhou, China. [3]Guangdong Province Engineering Laboratory for Medical Imaging and Diagnostic Technology, Guangzhou, China. [4]CardioCloud Medical Technology (Beijing) Co., Ltd., Beijing, China. [5]IT Department, Chinese PLA General Hospital, Beijing, China. [6]Department of Cardiology, Chinese PLA General Hospital, Beijing, China. ✉e-mail: fengqj99@smu.edu.cn; weiyanggm@gmail.com

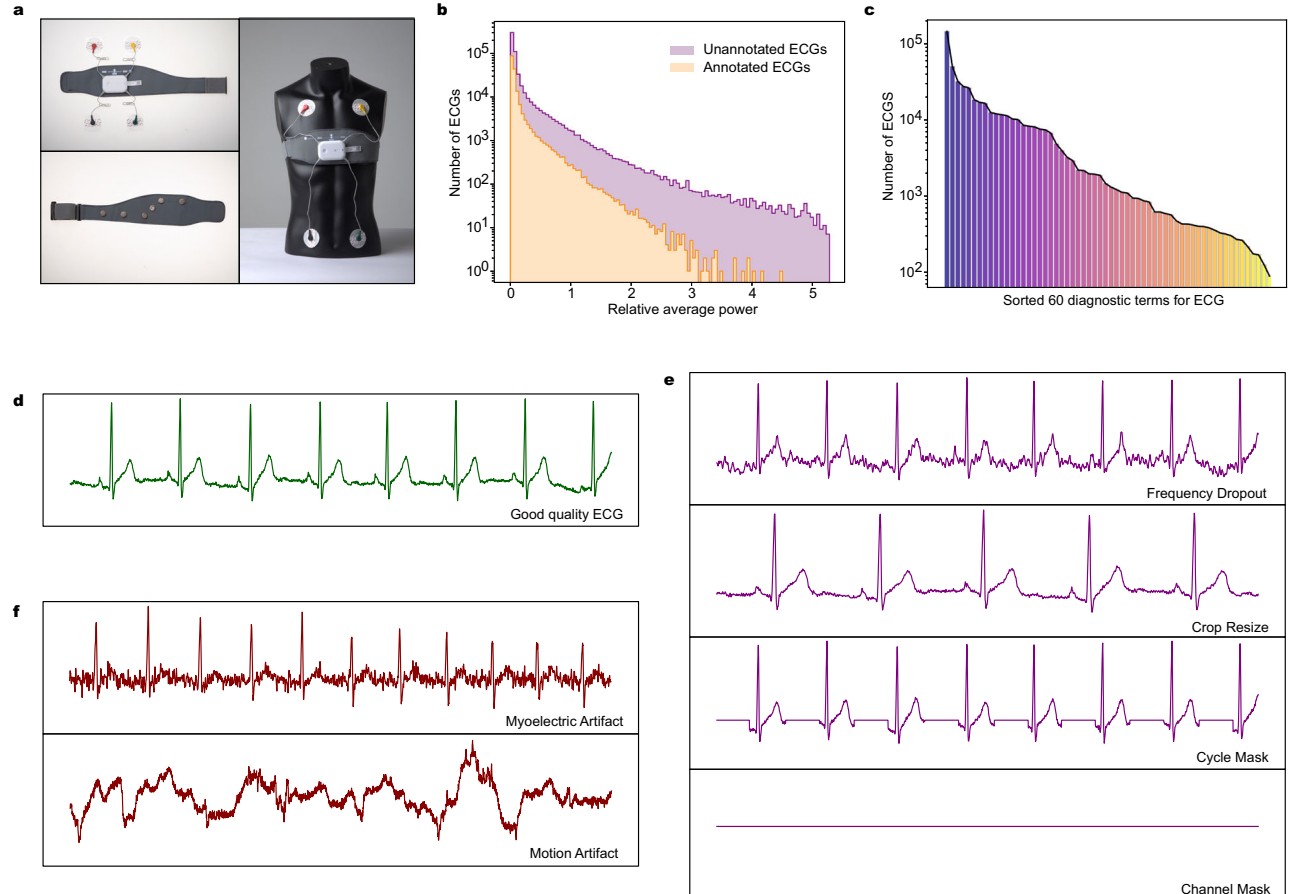

**Fig. 1 | Information for the large-scale ECG dataset. a** Wearable ECG device and its standard attachment method. Artifacts are formed in ECG due to electrode displacement and poor coupling to the skin in daily life. **b** Frequency distribution histogram of unannotated ECGs and annotated ECGs on relative average power. For ECGs of various power, unannotated data is always more than annotated data, and fully mining the knowledge of large-scale ECGs is expected to improve the classification performance of our model. **c** A typical long-tail distribution of 60 diagnostic terms for ECG, and the specific number of each class is shown in Table 1. **d** Example I lead of good-quality ECG signal with clear and smooth physiological waveform. **e** Four ECG augmentation operations based on above good-quality ECG. **f** Two most common artifacts in wearable ECG signals: myoelectric artifact (I lead) and motion artifact (V2 lead).

high-frequency noise; the motion artifact, also known as the baseline wander, appears as an unusual shift with large numerical amplitudes, and the electrode shedding signal is a straight line. The type of ECG studied by most of the current research is usually the standard 12-lead ECG used in hospitals, which is collected with the patient in a resting state and with little artifacts in signals. Cardiologists can make a diagnosis based on the clear part of an artifact-polluted ECG, however, the presence of artifacts in ECGs poses a challenge to computer-intelligent analysis. To improve the robustness of our model, we propose four data augmentation operations for 1D digital wearable multilead ECG signals, as shown in Fig. 1e, by simulating the interference that ECG faces in different scenes: random frequency dropout, random cycle mask, random crop resize and random channel mask.

Users upload large amounts of ECGs due to prolonged wearing of the device, and cardiologists usually diagnose a fraction of these data in combination with the patient's complaints, symptoms and time of onset, which generates a large amount of unannotated data. As shown in Fig. 1b, the distribution of unannotated ECGs completely includes the distribution of annotated ECGs, and there is no domain gap between them. Therefore, the reasonable utilization of these large-scale ECGs is expected to improve the classification performance of our model[2]. It is our motivation to use self-supervised learning to obtain knowledge valuable for the classification task by mining information about the data itself from large-scale ECGs. The method is to pre-train a network first, and then train the classification network

initialized by pretrained weights, rather than initialized randomly. The most common pretraining method is contrastive learning, which is generally based on the Siamese network architecture, to construct a pair of positive and negative samples to calculate the contrastive loss[3–5]. There are also methods based on image restoration, which train an encoder-decoder network to reconstruct pixel values of masked input data[6]. In general, when the amount of annotated data is small, a network initialized with pretrained weights can achieve better classification performance than a randomly initialized network[7].

Many research studies have achieved remarkable results in the field of automatic ECG analysis, including the detection and classification of arrhythmias[8–11], myocardial infarction[12,13], heart failure[14], hypertension[15] and sleep apnea[16]. These classification methods essentially consist of feature extraction and classification, and features before the era of deep learning mainly included physiological waveform features and statistical features. Physiological waveform features are used to describe the morphology attribute of heartbeats, such as the width, amplitude, slope, area and time interval of waveforms[17–19]. Statistical features are used to describe the quantitative and characteristic attributes of heartbeats, such as correlation[20], spectrum power[21], wavelet features[22,23] and high-order statistics[24]. Current recognition approaches often use deep features that are learnable and adaptive to the research object, describing ECG data more sufficiently compared with human-crafted features, and deep neural network in the field of intelligent ECG diagnosis has reached a level comparable to

human cardiologists[25]. Convolutional networks[26], recurrent networks[27] and Transformer[28,29] architectures are favored by many studies and widely used for arrhythmia classification. The most popular and largest open-source ECG dataset is PhysioNet/CINC2020[30], which includes five data sources with 6,877, 74, 516, 10,344 and 10,000 ECG recordings, respectively. The significant domain gap between the various sources is due to equipment and geography; many studies based on this dataset remain in the experimental research stage and lack the validation on large-scale ECGs from the real world[31–34].

In this study, we collected a total of 658,486 ECG recordings, among which 164,538 were annotated and the remaining 493,948 were unannotated. Each ECG in this study was annotated by two cardiologists and reviewed by three senior cardiologists to improve labeling consistency. To obtain a multiscale receptive field, we used multiscale convolution to improve the deep neural network[25] as backbone while reducing network parameters by half. We used self-supervised learning to mine the information contained in the massive ECGs, and transferred the learned knowledge to a classification task to better generalize unseen signals. We used data augmentation operations on the fly to improve the robustness of our model when training the network. Here, we report an approach to detect and recognize 60 diagnostic terms for ECG, including normal rhythm, sinus rhythm, 8 heartbeat waveform changes and 50 common arrhythmias, and our model maintained a high sensitivity and specificity in the additional 2-month online clinical test of 12,521 ECGs.

## Results
### Dataset
We established a large-scale, wearable 12-lead ECG dataset of 164,538 annotated ECGs and 493,948 unannotated ECGs from 76,482 individuals from 2016 to 2022. The duration of each recording was 15 s, and the digital sampling frequency was 500 Hz. The wearable device records the ECG according to the Mason-Likar 12-lead system, with a total of 10 electrodes, as can be seen in Fig. 1a. Four limb lead electrodes are placed on the upper-right shoulder, upper-left shoulder, lower-right chest, and lower-left chest to form leads I, II, III, AVR, AVL and AVF. Six chest lead electrodes are placed on the chest to form leads V1–V6.

The gold standard annotation procedure of the ECG dataset is as follows: each ECG was independently diagnosed and annotated by two cardiologists and subsequently reviewed by three other senior cardiologists who compared inconsistent annotations and submitted the final diagnosis. Difficult ECGs that are disputed or for which a definitive diagnosis cannot be given were submitted to an arbitrator for final review and determination. Among the 164,538 annotated ECGs, annotations made on 93,365 ECGs by senior cardiologists and the two cardiologists were identical and passed the review directly, while annotations of the remaining 71,173 ECGs, making up 43.26% of the total ECGs, were changed. This annotation process is complex and cumbersome, it was conducted only in this research project with the aim to make the model to be more reliable and the evaluation more objective and accurate. In the actual clinical setting, only one cardiologist provides the diagnosis.

Our ECGs were collected across China and cover all provinces, autonomous regions, and municipalities. A total of 67.3% of the dataset was comprised from male participants, and the remaining 32.8% were female. The ages of these 76,482 individuals ranged from 8 to 94 years, with an average age of 51 years. Because the anomalies in cardiac rhythms were not significantly associated with age or sex, we did not make additional considerations and designs for these factors. Of the total 164,538 ECGs, ~31,686 samples constituted healthy normal ECGs, with the remaining 132,852 ECGs exhibiting varying arrhythmia.

The task of recognizing and detecting these 60 diagnostic terms[35] for ECG are multi-labeled, and these cardiac rhythms are: normal ECG (NECG), sinus rhythm (SR), Q wave abnormal (QAb), q wave abnormal

(qAb), poor R wave progression (PRP), ST elevation (STE), ST depression (STD), T wave abnormal (TAb), peaked T waves (PT), elevated U wave (EU), Brugada Syndrome (BS), sinus tachycardia (ST), sinus bradycardia (SB), significant sinus bradycardia (SSB), sinus arrhythmia (SA), premature atrial contraction (PAC), atrial bigeminy (AB), atrial premature beat not transmitted (APNT), atrial premature beats with aberrations (APA), paired atrial premature beats (PAP), atrial escape beat (AEB), accelerated atrial escape rhythm (AAER), atrial fibrillation (AF), rapid atrial fibrillation (RAF), atrial fibrillation with intraventricular aberrant conduction (AFIVBC), atrial flutter (AFL), junctional escape (JE), junctional rhythm (JR), supraventricular tachycardia (SVT), atrial tachycardia (AT), premature ventricular contractions (PVC), paired ventricular premature beats (PVPB), ventricular bigeminy (VB), ventricular arrhythmia (VA), ventricular fusion beat (VFB), ventricular escape beat (VEB), ventricular tachycardia (VT), pre-excitation syndrome (PES), low limb lead voltage (LLLV), low chest lead voltage (LCLV), early repolarization (ER), first-degree atrioventricular block (IAVB), second-degree atrioventricular block-1 (IIAVB1), 3rd degree atrioventricular block (IIIAVB), bundle branch block (BBB), left anterior fascicular block (LAFB), complete right bundle branch block (CRBBB), incomplete right bundle branch block (IRBBB), complete left bundle branch block (CLBBB), partial right bundle branch block (PRBBB), nonspecific intraventricular conduction disturbance(NICD), 2nd sinoatrial block (IISAB), heart enlargement and hypertrophy (HEH), high left ventricular voltage (HLVV), high right ventricular voltage (HRVV), paced rhythm (PR), atrial paced rhythm (APR), ventricular paced rhythm (VPR), atrial-ventricular paced rhythm (AVPR), and atrial sensed ventricular paced rhythm (ASVPR). These classes obey a typical long-tail distribution as in Fig. 1c. These labels are two-level catalogs that parent catalog contains a certain number of secondary catalogs, and we placed the subordination in the Supplementary Data 1.

We have two test sets: the offline dataset[36] and the additional online dataset. Each user was assigned an identity document (ID) in our database, and the same user often uploaded several ECGs. We randomly divided the 164,538 ECGs into a training set with 157,538 samples and the offline test set with 7000 samples according to ID to ensure that the ECGs of the same individual corresponds to a single dataset. In addition, we deployed the network to the server and provided real-time artificial intelligence diagnosis for 12,521 ECGs uploaded by users who use the same wearable ECG device from January to February 2023. These recordings were also diagnosed by the cardiologist committee and all used for online testing. These ECGs conform to the distribution of patients in the real world, and positive samples of some classes are very rare, which allows more objective evaluation of the detection and classification performance of our approach.

### Metrics
We utilized two sets of evaluation indicators to evaluate the classification performance: label-based metrics and example-based metrics. Label-based metrics evaluating the classification performance of a model for each class separately in label or class wise: area under the receiver-operating characteristic curve (AUROC), area under the precision-recall curve (AUPRC), $Sen_{label}$, $Spe_{label}$ and $F1_{label}$. Example-based metrics evaluating the classification performance for each ECG in sample wise due to a single ECG may have multiple labels: $Sen_{exam}$, $Spe_{exam}$, $F1_{exam}$ and $Acc_{exam}$. AUROC measures the ability of the classifier to predict the sample correctly, and AUPRC comprehensive measures the precision and recall of positive samples. Sensitivity (Sen), also named recall, is the ratio of detected positive samples to all positive samples, and it is inversely proportional to the missed diagnosis rate of patients. Specificity (Spe) is the ratio of detected negative samples to all negative samples. F1 score is the harmonic average of recall and precision, and the accuracy (Acc) is the rate at which the model correctly classifies a sample. The value range of the above metrics is 0 to 1, and higher values indicate better performance.

## Experiment

We employed the improved multiscale ResNet18[25] as the backbone, using a total of 639,708 ECGs of all unannotated ECGs and the training set to pretrain a Siamese convolutional network via momentum contrastive learning, and then used the learned weights as the initialization weights of the downstream classification network. We used the combination of binary cross-entropy and pairwise ranking as the loss function for the multilabel classification task.

We conducted ablation experiments to analyze the respective contribution of pretrained weights, data augmentation operations and the multiscale convolutional layer to the classification performance in the proposed approach. Furthermore, we verified whether the pretrained weights outperformed the random weights using 10% to 100% of the training set, respectively, and found that the pretrained weights greatly improve the classification performance of our model when the data size is small. We implemented data augmentation on the fly using different hyperparameters to explore the contribution of different augmentation methods to the classification performance of the model. The offline test set was divided into a good-quality set and inferior-quality set according to ECG quality to analyze the impact of data augmentation on the robustness of the model. In addition, we analyze how the 12-lead based model can be extended to the common 1–3 lead ECG devices available on the market and the benefits that would be derived from doing so. We used the class activation map (CAM) to analyze the network's attention to the input ECG and to analyze whether the network has learned the knowledge of related arrhythmias. Finally, we tested our system on the China physiological signal challenge 2018[37] (http://2018.icbeb.org/Challenge.html, CPSC2018), and our F1 score was 0.839, which exceeded the best score published on their challenge website[38].

## Diagnostic performance

Our model achieved the average AUROC, average AUPRC and average F1 score of 0.975, 0.646 and 0.575, respectively, on the offline test set (Table 1). For the online test set, the average sensitivity, average specificity and average F1 score are 0.736, 0.954 and 0.468, respectively (Table 1). The automatic diagnosis result is visible to cardiologists only as a reference, and the final diagnosis has to be signed off by human experts. The distribution of ECG diagnostic terms in the online test set is more in line with the real-world distribution, and its positive samples are far rarer than those in the offline test set. In addition, ECGs in the online test set contains various artifacts due to daily use of the wearable device, while the data in the training set and offline test set are manually reviewed and choired for better data quality than the online test set, both of which result in some degradation in the model's performance when practical applied in real life, but still retain an acceptable level of sensitivity and specificity.

Intelligent diagnosis of ECGs in real clinical setting is a multi-label classification task, where a cardiologist may annotate multiple labels to a single ECG. Therefore, we evaluate the classification performance of the model in example wise. Our model was able to detect and classify 60 ECG diagnostic terms. As shown in Table 2, in the large-scale online test, each ECG had an average of 3.3 labels, and the intelligent diagnostic model was able to detect 2.7 of them, missing 0.6, while bringing in 2.5 false positives that were acceptable. This demonstrates that our model is able to diagnose most of the labels correctly, transforming the diagnostic task of cardiologists from a fill-in-the-blank to a multiple-choice and review, and can effectively reduce their workload.

The diagnosis of ECG is essentially the screening of possible cardiac diseases, and we are committed to detecting as many arrhythmic ECGs from the huge amount of data as possible to be diagnosed for further diagnosis by cardiologists. Although misdiagnosing negative recording as a false positive increases the time cost and workload of the cardiologist, a missed diagnosis of positive data wastes the opportunity for early treatment of patients, with very serious

consequences. For each cardiac rhythm, a threshold is required for the model to define whether the intelligent diagnostic result is positive. As shown in Fig. 2g, take the atrial fibrillation for example, the common recommended operating point on PR curve is the break even point or the optimal F1 point, but our senior cardiologists chose the fine-tuned point after considering recall and precision, where the recall is superior and the precision and F1-score are also acceptable. The F1 scores for both test sets are modest due to the large number of false positives generated by the severe data imbalance, but this is actually acceptable. As in the Table 3, even if the number of false positive cases is two, three or more times the number of true positive cases, this order of magnitude is insignificant compared to the large number of negative cases, and the specificity of 0.900 indicates that a large number of true negative ECGs are not misdiagnosed. In terms of sensitivity, the value of 0.795 indicates that most of the positive samples were detected. Therefore, our model can actually reduce the burden on cardiologists and has a great role and potential for real-time diagnosis and remote diagnosis.

## Ablation study

The combination of three strategies: pretraining weights (PW), multiscale convolution and data augmentation operations (Aug) work together in modeling. To study their effects, we carried out ablation on the offline test set. AUROC and AUPRC were calculated directly using the predicted values from the output of network and the ground truth, whereas sensitivity, specificity, and F1 score had to be calculated after the predicted values were binarized based on the preselected threshold. Among the two AUCs, AUPRC is more sensitive to data imbalance, so we used AUPRC as the major evaluation indicator for ablation analysis.

We designed five experiments to implement the multilabel classification task for 60 diagnostic terms for ECG: (i) *DNN*[22]: the 34-layer deep neural network (DNN) with random initialization, which is also our baseline; (ii) *MSDNN*: the improved 18-layer multiscale deep neural network (MSDNN) with random initialization; (iii) *MSDNN with PW*: fine-tuning the MSDNN initialized with the pretrained weights; (iv) *MSDNN with Aug*: using on-the-fly data augmentation operations, training the MSDNN with random initialization; (v) *MSDNN with Aug&PW*: using data augmentation operations, fine-tuning the MSDNN initialized with the pretrained weights.

The average AUPRC of these five experiments are 0.578, 0.582, 0.593, 0.637 and 0.646, respectively. As can be seen in Fig. 2a, when we used MSDNN instead of DNN as the backbone, the AUPRC improved by 0.4%. The lengths of the four parallel convolutional kernels in the multiscale convolutional layer are 3, 5, 9 and 17 instead of the fixed value of 17. This has two advantages: one is to obtain a multiscale receptive field, which is helpful for the analysis of subtle electrophysiological waveforms; the other is halving the number of parameters of each convolutional layer. With the network layers reduced from 34 to 18, the parameters of MSDNN are only 1/4 of DNN. The improvement in performance is not the main thing; the small-scale network provides technical support for future deployment to mobile terminals and can meet the demand of real-time monitoring and small computational resources. The AUPRC was improved by 1.1% using pretrained weights instead of random weight initialization. The features obtained from contrastive learning are derived from a very large ECG database, and using the knowledge contained in the ECGs themselves helps the network to generalize unseen data. The AUPRC of data augmentation was improved by 5.5%. We designed four data augmentation operations, two of which simulated the possible myoelectric artifact and lead-fail of wearable ECGs, and the remaining two operations randomly masked a fixed segment of each heartbeat and scaled ECG, respectively, which greatly improved the detection performance of the wearable ECG. When both data augmentation and pretrained weights are used, the classification performance is better

**Table 1 | Diagnosis performance of our model for 60 diagnostic terms for ECG using the offline test set and online test set**

| ECG Terms | Samples (n = 164538) | Offline test set (n = 7000) | | | Online test set (n = 12521) | | | | | | |
|---|---|---|---|---|---|---|---|---|---|---|---|
| | | AUROC | AUPRC | $F1_{label}$ | $TP_{label}$ | $TN_{label}$ | $FP_{label}$ | $FN_{label}$ | $Sen_{label}$ | $Spe_{label}$ | $F1_{label}$ |
| NECG | 31686 | 0.952 | 0.753 | 0.649 | 2134 | 9111 | 764 | 512 | 0.807 | 0.923 | 0.770 |
| SR | 142486 | 0.984 | 0.994 | 0.965 | 9200 | 2046 | 310 | 965 | 0.905 | 0.868 | 0.935 |
| QAb | 12437 | 0.953 | 0.746 | 0.774 | 347 | 11607 | 434 | 133 | 0.723 | 0.964 | 0.550 |
| qAb | 6705 | 0.935 | 0.509 | 0.528 | 51 | 11990 | 405 | 75 | 0.405 | 0.967 | 0.175 |
| PRP | 11947 | 0.906 | 0.510 | 0.460 | 446 | 11139 | 842 | 94 | 0.826 | 0.930 | 0.488 |
| STE | 7548 | 0.937 | 0.651 | 0.605 | 209 | 11233 | 867 | 212 | 0.496 | 0.928 | 0.279 |
| STD | 27266 | 0.934 | 0.799 | 0.722 | 917 | 10324 | 842 | 438 | 0.677 | 0.925 | 0.589 |
| TAb | 49719 | 0.960 | 0.908 | 0.828 | 2953 | 7700 | 1558 | 310 | 0.905 | 0.832 | 0.760 |
| PT | 1878 | 0.956 | 0.427 | 0.395 | 49 | 12339 | 106 | 27 | 0.645 | 0.991 | 0.424 |
| EU | 625 | 0.949 | 0.289 | 0.246 | 38 | 12263 | 131 | 89 | 0.299 | 0.989 | 0.257 |
| BS | 305 | 0.994 | 0.311 | 0.325 | 147 | 12037 | 251 | 86 | 0.631 | 0.980 | 0.466 |
| ST | 8086 | 0.990 | 0.896 | 0.790 | 325 | 11851 | 262 | 83 | 0.797 | 0.978 | 0.653 |
| SB | 18247 | 0.981 | 0.856 | 0.814 | 959 | 10530 | 853 | 179 | 0.843 | 0.925 | 0.650 |
| SSB | 405 | 0.990 | 0.517 | 0.523 | 86 | 12221 | 149 | 65 | 0.570 | 0.988 | 0.446 |
| SA | 16394 | 0.952 | 0.747 | 0.681 | 563 | 9591 | 2188 | 179 | 0.759 | 0.814 | 0.322 |
| PAC | 10042 | 0.985 | 0.896 | 0.840 | 1769 | 9815 | 681 | 256 | 0.874 | 0.935 | 0.791 |
| AB | 273 | 0.997 | 0.800 | 0.593 | 149 | 11573 | 746 | 53 | 0.738 | 0.939 | 0.272 |
| APNT | 946 | 0.967 | 0.420 | 0.446 | 152 | 11701 | 553 | 115 | 0.569 | 0.955 | 0.313 |
| APA | 1959 | 0.995 | 0.820 | 0.744 | 249 | 11855 | 333 | 84 | 0.748 | 0.973 | 0.544 |
| PAP | 936 | 0.987 | 0.553 | 0.419 | 329 | 10723 | 1420 | 49 | 0.870 | 0.883 | 0.309 |
| AEB | 1126 | 0.987 | 0.422 | 0.388 | 647 | 11107 | 434 | 333 | 0.660 | 0.962 | 0.628 |
| AAER | 623 | 0.995 | 0.646 | 0.508 | 440 | 11404 | 579 | 98 | 0.818 | 0.952 | 0.565 |
| AF | 8463 | 0.988 | 0.878 | 0.787 | 690 | 11368 | 379 | 84 | 0.891 | 0.968 | 0.749 |
| RAF | 1941 | 0.994 | 0.699 | 0.545 | 178 | 12093 | 230 | 20 | 0.899 | 0.981 | 0.587 |
| AFIVBC | 1103 | 0.991 | 0.679 | 0.652 | 68 | 12343 | 89 | 21 | 0.764 | 0.993 | 0.553 |
| AFL | 3961 | 0.996 | 0.977 | 0.928 | 30 | 12255 | 211 | 25 | 0.545 | 0.983 | 0.203 |
| JE | 873 | 0.985 | 0.712 | 0.617 | 518 | 10946 | 819 | 238 | 0.685 | 0.930 | 0.495 |
| JR | 360 | 0.991 | 0.494 | 0.352 | 140 | 11759 | 486 | 136 | 0.507 | 0.960 | 0.310 |
| SVT | 3225 | 0.989 | 0.789 | 0.675 | 205 | 11562 | 692 | 62 | 0.768 | 0.944 | 0.352 |
| AT | 2197 | 0.986 | 0.702 | 0.700 | 25 | 11733 | 753 | 10 | 0.714 | 0.940 | 0.062 |
| PVC | 8240 | 0.977 | 0.905 | 0.836 | 1490 | 10392 | 573 | 66 | 0.958 | 0.948 | 0.823 |
| PVPB | 172 | 1.000 | 0.830 | 0.232 | 78 | 11781 | 645 | 17 | 0.821 | 0.948 | 0.191 |
| VB | 221 | 0.995 | 0.376 | 0.282 | 224 | 11770 | 508 | 19 | 0.922 | 0.959 | 0.459 |
| VA | 434 | 0.892 | 0.178 | 0.102 | 215 | 10957 | 998 | 351 | 0.380 | 0.917 | 0.242 |
| VFB | 402 | 0.882 | 0.167 | 0.244 | 45 | 12024 | 357 | 95 | 0.321 | 0.971 | 0.166 |
| VEB | 128 | 0.982 | 0.350 | 0.142 | 86 | 11070 | 1349 | 16 | 0.843 | 0.891 | 0.112 |
| VT | 91 | 0.984 | 0.506 | 0.233 | 48 | 12239 | 223 | 11 | 0.814 | 0.982 | 0.291 |
| PES | 593 | 0.986 | 0.824 | 0.667 | 110 | 12199 | 162 | 50 | 0.688 | 0.987 | 0.509 |
| LLLV | 841 | 0.995 | 0.555 | 0.510 | 83 | 12181 | 210 | 47 | 0.638 | 0.983 | 0.392 |
| LCLV | 1474 | 0.990 | 0.598 | 0.547 | 22 | 12336 | 145 | 18 | 0.550 | 0.988 | 0.213 |
| ER | 2925 | 0.957 | 0.454 | 0.449 | 34 | 12204 | 236 | 47 | 0.420 | 0.981 | 0.194 |
| IAVB | 11633 | 0.973 | 0.747 | 0.715 | 418 | 11414 | 499 | 190 | 0.688 | 0.958 | 0.548 |
| IIAVB1 | 321 | 0.996 | 0.712 | 0.600 | 73 | 12012 | 416 | 20 | 0.785 | 0.967 | 0.251 |
| IIIAVB | 177 | 0.998 | 0.781 | 0.539 | 83 | 11652 | 753 | 33 | 0.716 | 0.939 | 0.174 |
| BBB | 26093 | 0.965 | 0.891 | 0.849 | 1610 | 9485 | 1108 | 318 | 0.835 | 0.895 | 0.693 |
| LAFB | 1319 | 0.994 | 0.539 | 0.571 | 515 | 11531 | 425 | 50 | 0.912 | 0.964 | 0.684 |
| CRBBB | 4844 | 0.985 | 0.831 | 0.729 | 629 | 11612 | 225 | 55 | 0.920 | 0.981 | 0.818 |
| IRBBB | 11294 | 0.961 | 0.744 | 0.745 | 358 | 11509 | 529 | 125 | 0.741 | 0.956 | 0.523 |
| CLBBB | 1224 | 1.000 | 0.955 | 0.839 | 85 | 12366 | 45 | 25 | 0.773 | 0.996 | 0.708 |
| PRBBB | 7388 | 0.910 | 0.442 | 0.469 | 17 | 11830 | 654 | 20 | 0.459 | 0.948 | 0.048 |
| NICD | 332 | 0.979 | 0.281 | 0.281 | 118 | 11871 | 499 | 33 | 0.781 | 0.960 | 0.307 |
| IISAB | 266 | 0.995 | 0.217 | 0.192 | 63 | 11952 | 486 | 20 | 0.759 | 0.961 | 0.199 |
| HEH | 16865 | 0.935 | 0.728 | 0.671 | 777 | 10908 | 538 | 298 | 0.723 | 0.953 | 0.650 |
| HLVV | 10231 | 0.975 | 0.829 | 0.752 | 727 | 11279 | 432 | 83 | 0.898 | 0.963 | 0.738 |

**Table 1 (continued) | Diagnosis performance of our model for 60 diagnostic terms for ECG using the offline test set and online test set**

| ECG Terms | Samples (n = 164538) | Offline test set (n = 7000) | | | Online test set (n = 12521) | | | | | | |
|---|---|---|---|---|---|---|---|---|---|---|---|
| | | AUROC | AUPRC | $F1_{label}$ | $TP_{label}$ | $TN_{label}$ | $FP_{label}$ | $FN_{label}$ | $Sen_{label}$ | $Spe_{label}$ | $F1_{label}$ |
| HRVV | 421 | 0.995 | 0.430 | 0.444 | 87 | 11961 | 459 | 14 | 0.861 | 0.963 | 0.269 |
| PR | 2170 | 0.994 | 0.916 | 0.901 | 561 | 11874 | 55 | 31 | 0.948 | 0.995 | 0.929 |
| APR | 435 | 0.982 | 0.769 | 0.783 | 26 | 12465 | 23 | 7 | 0.788 | 0.998 | 0.634 |
| VPR | 572 | 0.998 | 0.693 | 0.681 | 267 | 12076 | 159 | 19 | 0.934 | 0.987 | 0.750 |
| AVPR | 475 | 1.000 | 0.971 | 0.898 | 85 | 12251 | 182 | 3 | 0.966 | 0.985 | 0.479 |
| ASVPR | 386 | 0.991 | 0.155 | 0.103 | 136 | 12211 | 171 | 3 | 0.978 | 0.986 | 0.610 |
| **Average** | 8095.5 | 0.975 | 0.646 | 0.575 | 551.4 | 11327.2 | 523.9 | 118.6 | 0.736 | 0.954 | 0.468 |

**Table 2 | Diagnosis performance of our model in example wise using the offline test set and online test set**

| Test set | $TP_{exam}$ | $TN_{exam}$ | $FP_{exam}$ | $FN_{exam}$ | $Sen_{exam}$ | $Spe_{exam}$ | $F1_{exam}$ | $Acc_{exam}$ |
|---|---|---|---|---|---|---|---|---|
| Offline (n = 7000) | 3.0 | 56.1 | 1.4 | 0.5 | 0.873 | 0.975 | 0.756 | 0.969 |
| Online (n = 12521) | 2.7 | 55.3 | 2.5 | 0.6 | 0.857 | 0.956 | 0.653 | 0.950 |

than using pretrained weights alone, but not significantly improved compared to using data augmentation alone.

## Comparison of initialization by pretrained weights and random weights

For deep learning approaches, the larger the amount of ECGs, the more reliable the classification performance of the model. Cardiologists are also concerned about how much ECG data is needed to make the model practical. In order to analyze the advantages of pretraining weights, we randomly selected 10–100% of training ECGs as sub-training sets in turn, and initialized the classification network with pretraining weights and random weights, respectively, to observe whether the self-supervised learning method could improve the classification performance of the model. As the amount of ECG increased, the two curves in Fig. 2b converged to 0.582 and 0.593, respectively. Based on the trend, the classification performance of our model is limited primarily by the difficulty of the classification task itself rather than by a lack of data, and it is unlikely that there will be a significant improvement in intelligent diagnostic performance even with additional ECGs. The improvement in classification performance from pretraining was most significant at 10% of the annotated ECGs, with AUPRC improving by 3.1%. The benefits of pretraining diminish as the amount of data rises, decaying to 1.1% when using all annotated ECGs. This suggests that our current amount of ECGs is sufficient to make the model practical and that the self-supervised approach contributes to improvement of the model generalizability.

## ECG augmentation and model robustness

We trained models using four different sets of ECG augmentation hyperparameters and compared the results with their combination. As shown in Fig. 2c, each data augmentation operation is effective in improving the AUPRC, and their combination has the most significant improvement in classification performance, a 5.5% improvement in AUPRC compared to the baseline. In addition, to investigate whether data augmentation improves the robustness of the model, we divided the offline test set into a good-quality set and an inferior-quality set according to the signal quality with 5,209 and 1,791 ECGs, respectively. As can see in Fig. 1d and Fig. 1f, the ECG signal in the good-quality set is clear and smooth, while most ECGs of the inferior-quality set have severe myoelectric artifacts, motion artifacts and lead electrode dislodgement, which can even affect the cardiologist's diagnosis. Overall, the AUPRC is highest for the good-quality set, intermediate for the offline test set, and worst for the inferior- quality set as in Fig. 2d, which indicates that the presence of interference or artifacts in the signal

does affect the classification performance of the model. The improvement in AUPRC using data augmentation is 5.5% on the offline test set, 4.9% on the good-quality set, but 7.1% on the inferior-quality set, which indicates that ECG augmentation is effective in improving the robustness of the model, making it quite resistant to the presence of artifacts in the input signal.

## Extending the model based on 12-lead ECG to 1–3 lead ECG devices

Most of the widely used wearable ECG devices on the market are 1–3 leads, while the object of our study is 12-lead wearable ECG signals. In order to extend the applied range of our model to these 1–3 lead ECG devices, we propose a feasible solution: using data generation methods such as generative adversarial networks (GAN) to transform 1–3 leads ECG into the currently used 12 leads ECG, which then can be straightforward to use our model. However, we currently do not have mature 1–3 leads ECGs dataset and corresponding annotation for study.

To explore possible benefits of such behavior, we used 1–3 of the 12 leads to simulate the signals acquired from 1 to 3 wearable ECG devices: using lead I to simulate the single-lead device; using II, V1 and V5 to simulate the 3-lead Holter ECG device used in the hospital; using I, AVF and V2 to simulate the 3-lead Frank orthogonal lead ECG device. Notably, in these three simulation scenarios really 1–3 leads are only similar to the listed sub-leads in waveforms, albeit they are not part of any of the 12 leads. We compared the classification performances of these three cases with the 12-lead ECG as Fig. 2e, yielding the average AUPRC are 0.464, 0.586, 0.584 and 0.646, respectively. The single-lead signal is significantly less informative than the 3-lead and 12-lead, and the 3-lead ECG also not as informative as the 12-lead ECG. We consider that the higher the number of leads would yield to the observation of richer information and yield more accurate the diagnostic results. This approach obtains the intelligent diagnostic results on the level of the 12-lead ECG device with a lower cost and more convenient device.

## Model attention visualization

We used the class activation map (CAM) to analyze our model's attention to the input ECGs and whether the network has learned the knowledge of related ECG diagnostic terms. We observed that the parts of the model that focused on were consistent with those of human cardiologists. As shown in Fig. 3c, the part circled in purple dashed line is the heartbeat related to arrhythmia defined by human experts, and the model paid the highest attention to this part.

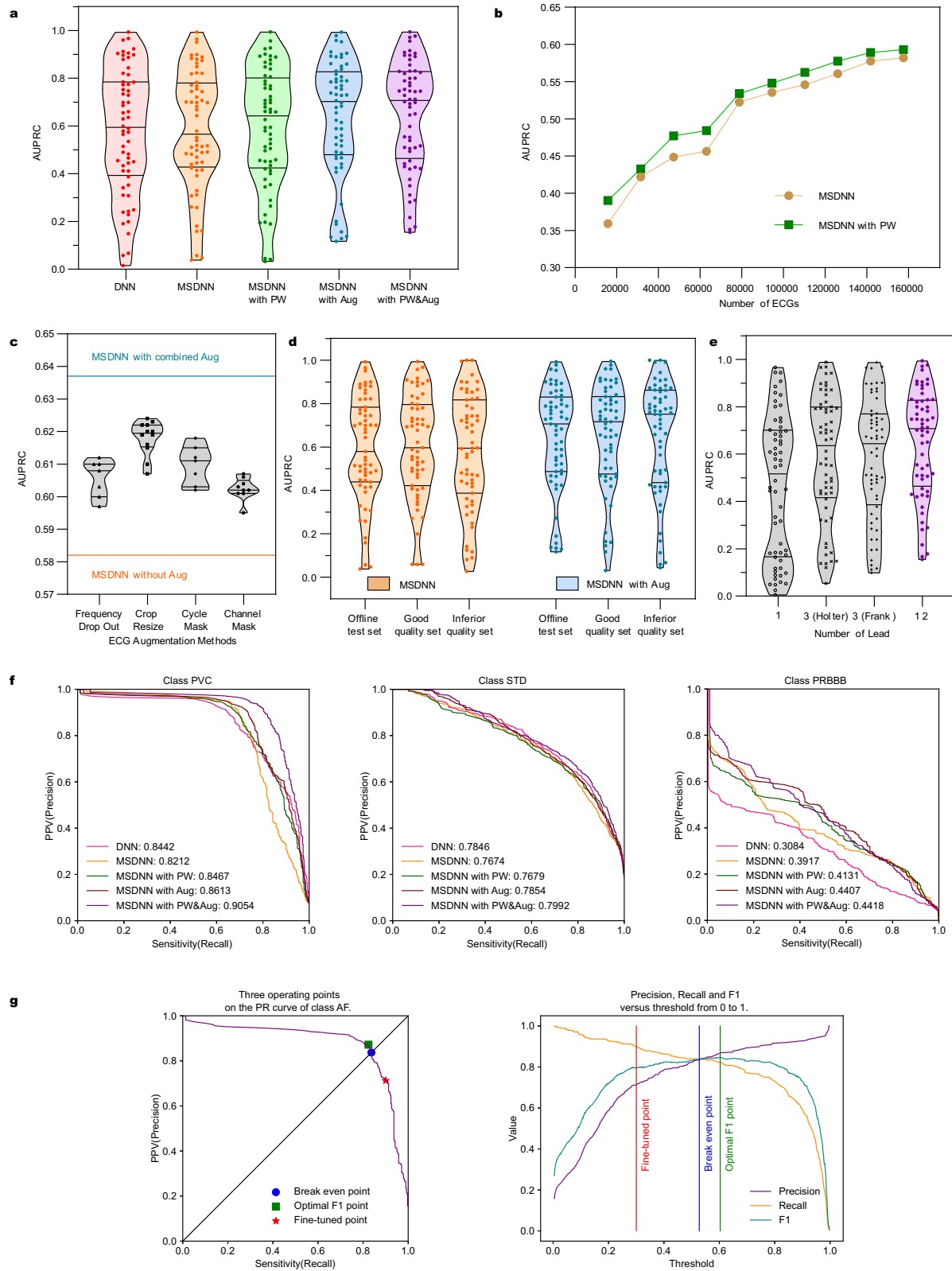

## Open-source dataset evaluation

To validate the effectiveness and generalization of the proposed approach to external data, we conducted the experiment on the China physiological signal challenge 2018 dataset[37] (http://2018.icbeb.org/Challenge.html, CPSC2018). The challenge ECG recordings are 12-lead standard ECG with time length from 6 s to 60 s collected from 11 hospitals. Its training set includes 6,877 ECGs available to the public,

and its unavailable test set includes 2,954 ECGs. The dataset contains nine cardiac rhythms or annotations: normal, atrial fibrillation (AF), first-degree atrioventricular block (I-AVB), left bundle branch block (LBBB), right bundle branch block (RBBB), premature atrial contraction (PAC), premature ventricular contraction (PVC), ST-segment depression (STD) and ST-segment elevation (STE), and uses average F1 as the evaluation indicator. We employed 10-fold cross-validation to

**Fig. 2 | The performance analysis, threshold selection and extension of application of the model. a** Ablation study of three strategies: pre-training weights (PW), multi-scale (MS) convolution and data augmentation operations (Aug). **b** Improvement of classification performance initialized by pre-trained weights compared to random weights on different number of training sets. **c** Improvement of classification performance by ECG augmentation operations. Each augmentation operation has a different degree of improvement, and the combination of them has the most significant improvement in classification performance. **d** Effect of

classification performance from ECG augmentation on offline test set, good-quality set and inferior-quality set. The offline test set is divided into good-quality set and inferior-quality set according to the data quality, and the improvement of classification performance from data augmentation was most pronounced on the inferior-quality set, indicating that our model is robust to the artifacts in ECGs. **e** Classification performance when only partial leads are used. **f** Precision-recall curves of IAVB, AF and STE. **g** The operating point selection of class AF on the PR curve. Source data are provided as a Source Data file.

**Table 3 | Confusion matrix, F1, sensitivity, precision and specificity at different thresholds of atrial fibrillation**

| Threshold | TP | TN | FP | FN | F1 | Sensitivity | Precision | Specificity |
|---|---|---|---|---|---|---|---|---|
| 0.004 | 380 | 4592 | 2028 | 0 | 0.273 | 1.000 | 0.158 | 0.694 |
| 0.100 | 364 | 5984 | 636 | 16 | 0.528 | 0.958 | 0.364 | 0.904 |
| 0.201 | 352 | 6373 | 247 | 28 | 0.719 | 0.926 | 0.588 | 0.963 |
| **0.300** | **342** | **6482** | **138** | **38** | **0.795** | **0.900** | **0.713** | **0.979** |
| 0.400 | 328 | 6531 | 89 | 52 | 0.823 | 0.863 | 0.787 | 0.987 |
| **0.528** | **318** | **6558** | **62** | **62** | **0.837** | **0.837** | **0.837** | **0.991** |
| **0.603** | **313** | **6574** | **46** | **67** | **0.847** | **0.824** | **0.872** | **0.993** |
| 0.704 | 293 | 6587 | 33 | 87 | 0.830 | 0.771 | 0.899 | 0.995 |
| 0.802 | 277 | 6595 | 25 | 103 | 0.812 | 0.729 | 0.917 | 0.996 |
| 0.900 | 225 | 6608 | 12 | 155 | 0.729 | 0.592 | 0.949 | 0.998 |
| 0.971 | 109 | 6612 | 8 | 271 | 0.439 | 0.287 | 0.932 | 0.999 |
| 0.998 | 1 | 6620 | 0 | 379 | 0.005 | 0.003 | 1.000 | 1.000 |

The three bold lines from top to bottom are the thresholds, confusion matrixs and classification performance when selecting the fine-tuned point, the break even point and the optimal F1 point respectively. Our principle in selecting the operating point is higher sensitivity and fewer false negative samples, but at the same time it will lead to an increase in false positive samples. Even though the false positives of the fine-tuned threshold (138) are double or more than the false positives of the default threshold (62), which will increase the workload of cardiologists, but it is worth it for the access it gives patients.

train networks and used simple voting for ensemble, and our F1 score is 0.839, which exceeded the best score published on the challenge website 0.837[38]. This shows that our approach is not only suitable for the ECG signals collected by the proximal limb leads of a wearable device; it also can generalize the analysis tasks to the standard 12-lead ECG signals collected in hospital by the distal limb leads.

## Discussion

This study used large-scale ECGs to learn the physiological waveform knowledge contained in the data itself via contrastive learning, embedding these recordings into a low-dimensional manifold to obtain mutually exclusive distribution as the initialization state for related classification tasks. For stronger induction ability, we used multiscale convolution to improve the baseline deep neural network. Our four data augmentation operations designed for 1D digital ECG signals effectively improved the classification performance and robustness of the model. The parameter of our network is only 2.13 M, and the predict time cost for one ECG recording is within 0.08 s, which means that our model can be easily deployed to mobile devices and servers, and it can serve the public in real time, with small computing resource requirements. In addition, our model has achieved the confirmation of cardiologists in online testing and can provide users with real-time intelligent diagnosis and long-term continuous online automatic monitoring, which can practically reduce the workload of cardiologists.

In the 2-month online test, the average sensitivity of our model was 0.736, which means that 26.4% of positive ECGs were missed, which could lead to patients missing the precious immediate access to a cardiologist. While the average specificity is good at 0.954, which means that 4.6% of the ECGs were false alarms. Falsely alerted ECGs may cause psychological burden to the collector and increase the workload of the cardiologist. The performance of model is limited by the inherent difficulty of the task itself and the power of algorithm and

computer, we must weigh the sensitivity and specificity by choose a suitable operating point in the context of allowing the model to make mistakes. Sensitivity and specificity are highly correlated with the choice of the operating point, with sensitivity increasing and specificity decreasing as the threshold decreases. In this regard, we will design specific strategies and methods to the characteristics of the wearable 12-lead ECG and the problems faced in clinical using to improve the classification performance of the model. The average F1 is modest at 0.468, mainly because the optimal F1 point was not selected as the operating point and there is a severe data imbalance in our dataset. The selection of operating points is manually fine-tuned by senior cardiologists, and we will study the automatic threshold selection method. Data imbalance is also a common difficulty in medical data analysis, our current approach to mitigating it is class weighting, and we will use more refined methods to cope with this hindrance in future work.

At the algorithm level, even if we simulate several kinds of interference that wearable ECGs may encounter in certain scenes, our efforts are far from enough due to more complicated predicaments in the real environment encountered by wearable ECG devices. During the online test, we found that our model was mainly missing and misdiagnosing the ECG data that were contaminated by interference, and we will study the decoupling of signal and artifacts in ECG with interference in the future, and coupling the ECG with artifacts decomposed from other ECG as a data augmentation operation to further improve the classification performance of the model. In addition, the classification performance of the self-supervised learning framework can be further improved, and we will combine semi-supervised learning methods to further mine the knowledge contained in the ECG signal itself and improve the generalization performance of the model. Finally, there are symbiotic and mutually exclusive relationships between multiple fine-grained labels, which is the advantage and also the challenge of our large-scale dataset, and we should apply

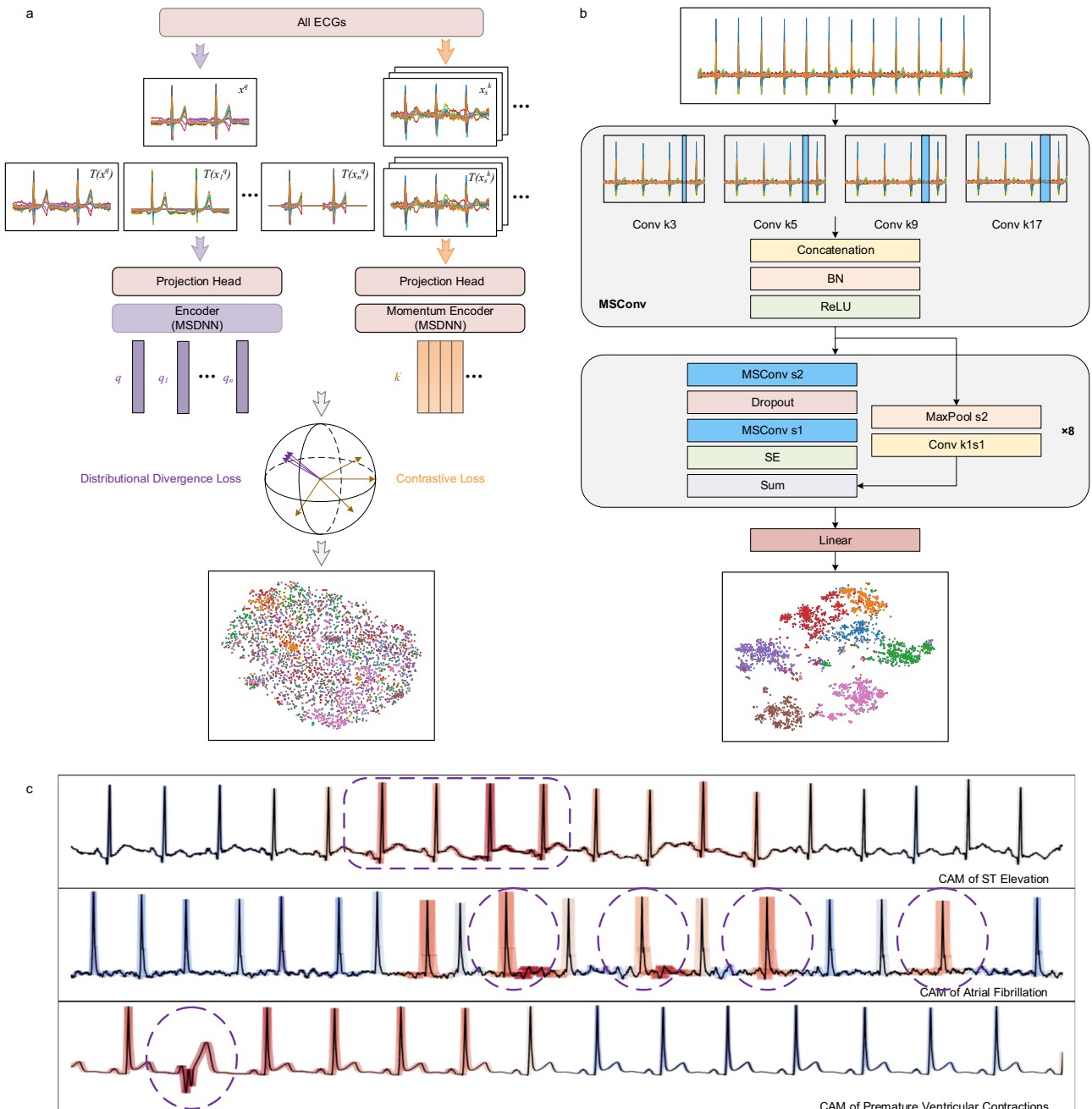

**Fig. 3 | Neural network architectures and visualization of model's attention.**
**a** The improved MoCo architecture for pretraining. **b** The architecture of our multiscale convolutional network. **c** Attention visualization of our model to input ECG on three arrhythmias using CAM, the red background indicates where the model pays more attention. The purple circle indicates the human expert's definition of this arrhythmia.

this a priori knowledge to the method to obtain more reasonable and natural diagnostic results.

Although deep learning has been successfully applied to physiological signal processing, our model can be initially applied to clinical diagnostic settings. However, there is significant variation in ECGs due to collected from different devices, geographic areas, skin tones and muscle mass etc., our model is currently only applicable to the population using the specific wearable ECG devices within China. In addition, the number of classes in our database close to 300, many of which only haves few samples such as ventricular tachycardia (VT) whose sample size is <100. Currently, the lack of samples is the main factor limiting the classification performance of the model to rare classes. In future work, we will expand our data sources to collect ECGs from different ethnic groups, populations and devices, and try to

bridge the domain gap between the distribution of ECGs from different sources, try our best to recognize more diagnostic terms for ECG at a fine-grained level and extend this work to a wider range of applications. In conclusion, we have established a baseline as far as the practicability of deep learning approach, which can provide a reference and feasible report for other researchers. The difficulties encountered by the model in real-world settings are more complex and there is much more work we can study, which provides more opportunities and challenges for the analysis and processing of ECG.

## Methods
### Data access and ethical statement
Samples for this study were obtained from three sources including home users wearing the wearable devices to collect ECG at home,

patients in hospitals who used wearable devices, and companies and the hospital organizing free ECG diagnostic events to collect ECG data. It is noteworthy that the ownership of the data belongs to Cardiocloud Medical Technology (Beijing) Co., Ltd. This study was approved by the Medical Ethics Committee of Chinese PLA General Hospital with the number of S2019-318-03, and informed consent was obtained from the participants.

## Pre-filtration of ECGs

Two factors may affect the quality of ECG signal in wearable ECGs, namely lead reversal and artifacts. Lead reversal occurs due to lead electrodes of wearable ECG devices being misplaced on the skin due to unfamiliarity during use by non-medical professionals, such as home users. Artifacts are caused by bad contact or even shedding between the lead electrodes and the skin. These two factors can lead to significant distribution differences between the data collection and normal wearing situations, which severely decreases the classification performance of the model. Our solution is to use the deep learning model to correct the wearing situation of users before uploading ECG, and to exclude the ECG data acquired in an inappropriate wearing situation.

There are six types of device wearing situations: upper limb left and right lead reversal, upper and lower limb lead reversal, left upper and lower limb lead reversal, right upper and lower limb lead reversal, chest lead reversal, and normal wear connection. There are total of 230 lead reversal ECGs in our annotated 164,538 ECGs, all of which are diagnosed by experienced human cardiologists. A domain gap exists between the distribution of lead reversal ECGs and normal wearing ECGs; however, due to the small amount of data, we are currently unable to accurately analyze the impact of this difference on the model classification performance. We used the MSDNN as the backbone to train a six-class network to identify the six wearing situations using cross-entropy as the loss function, and deployed it to mobile devices as a startup detection. We recruited 95 volunteers and collected 575 ECGs with a length of 15 s. The training set consisted of 482 ECGs from 80 of these individuals, and the test set consisted of 93 ECGs from the remaining 15 individuals. Of the 93 data tested, only one case of left upper and lower limb lead reversal was misdiagnosed as upper and lower limb lead reversal, while the rest of the data were detected correctly.

To evaluate artifacts, it is necessary to determine whether the ECG is acceptable via quality assessment We use the deep learning approach to assess the signal quality of each lead, and filter out ECGs of bad quality that do not have diagnostic value before inputting the data into the classification model. This method is described in detail in our previous study[39].

These two works have three directions of future improvement: recruiting more volunteers to collect richer data; reducing the detection time from 15 s to shorter such as ideally 1 s, correcting the wear status instantly when the user starts collecting signal; and light-weighting the model for deployment on mobile devices.

## Data preprocessing

More than 90% of the energy in ECGs is concentrated between 0.5–35 Hz, and this frequency band contains most of the diagnostic information from the signal. There are three main types of ECG interference: power frequency interference, generally at 50 Hz or 60 Hz; myoelectric interference, generally at 30 to 300 Hz; and baseline wander, generally <0.5 Hz. Benefiting from the electromagnetic safety design of the wearable device hardware, there is basically no power frequency interference in our dataset. Overall, low-frequency baseline wander is detrimental to model training, while middle- and high-frequency components should be preserved. Therefore, we pre-processed the data with a 5th order Butterworth high-pass filter, with the lower cutoff frequency of 0.5 Hz.

## Data augmentation

Data augmentation in the field of computer vision[4,40] mainly includes color transformation, scaling, affine transformation and masking, which can significantly improve the accuracy of the model, but there are relatively few studies involving data augmentation in the field of electrocardiogram analysis[41–43]. Our data augmentation operations not only simulate the various interferences faced by the ECG in the real world and improve the robustness of the model, but also direct more attention to the key features of the input signal through randomness and play a regularization role to suppress model overfitting. Accordingly, we proposed four data augmentation operations for wearable ECG signals: (1) Frequency dropout: transform the ECG signal to frequency domain via discrete cosine transform (DCT), then a certain number of frequency components are randomly set to zero; finally, the signal is transformed to the time domain by inverse discrete cosine transform (IDCT). (2) Crop resize: randomly crop a segment of the signal in the ECG data and resample it to the original length, and the crop position is randomly selected. (3) Cycle mask: detect the position of the R wave peak in ECG and set a segment of signal at the same position in each heartbeat to zero. (4) Channel Mask: randomly select a few channels to reset their signal value to zero. Figure 1d, e compare the original signal and the four data augmentation operations.

## Algorithm development

The annotation of ECGs is diagnosed by multiple cardiologists, and it is expensive and time-consuming because of the medical knowledge involved. Due to the relative scarcity of medical resources, unannotated ECG signals are easier to obtain than annotated ECG signals. The self-supervised learning can significantly improve the classification accuracy of the model in the field of computer vision, and it is our motivation to mine the information in the large-scale ECG data to improve the generalization of the classification model.

Momentum Contrast (MoCo)[3] is mainly composed of an encoder and a momentum encoder, and there is a projection head at the bottom of each encoder to increase the nonlinearity of hidden representation projected into the low-dimensional manifold. It stores a fixed number of representations or keys as the dictionary and calculates the contrast loss between the query and keys. MoCo updates the encoder by the gradient of contrast loss, and then uses the encoder weights to update the momentum encoder weights using momentum moving average, to maintain the consistency between the query and keys in the dictionary. This Siamese network embeds all recordings into a low-dimensional manifold of a specified dimension like 128, treats each ECG recording as a separate class, and uses the contrastive loss to maximize the interclass divergence. In this low-dimensional manifold, the distribution of different augmented representations of the same anchor ECG should be similar. Therefore, we improved MoCo with the distribution divergence loss[4] to minimize the intraclass divergence.

Formally, as shown in Fig. 3a, for an anchor ECG recording $x$, we randomly applied ECG data augmentation $T$ to get N+1 different views: $T(x^q)$, $T(x_1^q)$, $T(x_2^q)$ until $T(x_n^q)$, and their encoded representations are $q$, $q_1$, $q_2$, ... and $q_n$, respectively. The dictionary $k^-$ is defined as a buffer tensor including K representations, and all of them are negative keys. We denote $q$ as $k^+$ and concatenate with $k^-$ to get a tensor $k$ with K+1 representations, and then use a (K+1)-way softmax-based classifier to calculate the contrastive loss $L_C$ as Eq. (2). We can obtain a joint probability distribution $P(q,k)$ as Eq. (1) of $q$ and $k$, and these distributions of $P(q_1,k)$, $P(q_2,k)$, ... and $P(q_n,k)$ should be similar to $P(q,k)$, then we calculate the Kullback–Leibler divergence as distribution divergence loss $L_D$ as Eq. (3). The contrastive loss $L_C$, distribution divergence loss $L_D$ and the total loss $L$ are defined as:

$$P(q,k) = \frac{\exp(q.k^+/\tau)}{\exp(q.k^+/\tau) + \sum_{i=1}^{K}\exp(q.k^-/\tau)} \quad (1)$$

$$L_C = -\log[P(q,k)] \tag{2}$$

$$L_D = \sum_{i=1}^{N} P(q,k) \log\left[P(q,k)/P(q_i,k)\right] \tag{3}$$

$$L = L_C + \beta L_D \tag{4}$$

where $\beta$ is the weight coefficient of distribution divergence loss with typical value of 0.15, and N is the number of augmentation views, with a typical value of 3.

### Network architecture

We reproduced the 34-layer convolutional network[25] for comparison and employed it as our baseline. There are two slight changes: For convenience of padding computation, we used the convolution kernel with a width of 17 instead of 16; we also added the squeeze-and-excitation block to each convolutional module to implement the channel attention mechanism. The loss function of the multilabel classification task is the combination of binary cross entropy and pairwise ranking loss[44], weighted by the inverse of the number of positive recordings for each class.

As shown in Fig. 3b, convolution, batch normalization and pooling are all 1-D operations. "Conv" represents a convolutional layer, "k" is the width of the convolution kernel, and "s" is the stride of the convolution. "MSConv" is a multiscale convolution block, which uses four parallel convolution kernels with kernel widths of 3, 5, 9 and 17 to replace the common convolution kernel of width 17, finally concatenating feature maps channelwise. When the number of channels of MSConv is N, the number of channels of each parallel group convolution is N/4. Each residual module includes two MSConv blocks as a down-sampling module, and there are eight such residual modules in total. The number of channels of the convolution kernel in each residual layer is $64 + 16k$, where k is a hyperparameter, starting from 0 and incrementing by 1 for each residual block, with a dropout rate of 0.2. MSDNN is a multiscale ResNet18 whose input is a 1-D 12-channel ECG signal with a length of 15 seconds, and the output is a predicted probability vector activated by the sigmoid function. The parameters of DNN and MSDNN are 2.03 M and 8.03 M, respectively.

Our experiment was implemented on Pytorch 1.10[45]. We pre-trained the Siamese network on all ECGs with 100 epochs and used the Adam optimizer with the default parameters at the learning rate of 0.001, $\beta_1$ of 0.9 and $\beta_2$ of 0.999. The improved MoCo has a momentum of 0.9, a feature dimension of 128, a softmax temperature of 0.07, a queue size of 72,000 and a batch size set to 360. When training classification tasks, we used the SGD optimizer with momentum of 0.9 to minimize the loss function and set the mini-batch size to 128 and weight decay to $5e^{-4}$. When the network was initialized randomly[46], the learning rate schedule is one cycle with 100 epochs. The learning rate starts from 0.001 and increases to 0.01 at the 45th epoch (this part is called warm-up); then decays to 0.001 at the 90th epoch and to $1e^{-6}$ during the last 10 epochs. When the network is initialized as pretrained weights, the learning rate schedule only uses the latter half of the one cycle without warm-up. We saved the model with the lowest loss on the test set.

### Operating point selection

Our model currently detects and recognizes 55 cardiac rhythms. A single ECG may have more than one label and is classified by multi-label classification task. The activation function of network's final layer is sigmoid and the loss function is a combination of binary cross-entropy and pairwise ranking loss. The model has 55 prediction values ranging between 0 and 1 for each ECG to be diagnosed. To obtain the final result 55 thresholds are required to binarize these prediction values, with those greater than or equal to the threshold being 1 and those less than the threshold being 0. The thresholds vary for each cardiac rhythm and are fine-tuned by our senior cardiologists from the default thresholds after taking into account the sensitivity and specificity of each class.

Figure 2g and Table 3 show that for the current cardiac rhythm (AF), upon selecting the operating point on the PR curve of offline test set, the threshold of recommended operating point is the break even point of 0.389, where the precision, recall and F1 score are all equal to 0.746. Break even point is not the optimal F1 point, and the F1 score peaks at 0.748 corresponding to a threshold of 0.282. However, both sensitivities at these two operating points are below 0.8, and a lower threshold and higher sensitivity are preferred to reduce the number of missed positives. Nevertheless, this would multiply the number of false positives and increase the workload of cardiologists. Our senior cardiologists finally selected a threshold of 0.1 after systematic consideration, yielding a sensitivity and F1-score of 0.887 and 0.718 respectively. At the same time the false positives of the fine-tuned threshold (767) are double or more than the false positives of the default threshold (334). A single ECG review takes the cardiologist 3–5 min generally, but the added expense and cost is well worth it compared to the cost of a missed diagnosis.

### Statistical analysis

Among the all evaluation indicators we used, AUPRC comprehensively considers recall and precision, which can most objectively evaluate the classification performance of different models. Therefore, we performed paired-sample $t$-tests on the following sets of results to identify whether different strategies have a significant effect on the improvement of classification performance: DNN vs MSDNN, MSDNN vs MSDNN with PW, MSDNN vs MSDNN with Aug, and MSDNN vs MSDNN with PW when only 10% training set ECGs was used; the $p$-values were 0.749, 0.202, 0.000 and 0.015, respectively. In terms of network architecture, there is no significant difference in classification performance between DNN and MSDNN, but the number of parameters of MSDNN is only 1/4 of that of DNN. The ECG augmentation strategy improves the model classification performance significantly. The strategy of pretrained weights showed a significant improvement in classification performance when the data scale was small, but its significance gradually decreased with the gradual increase of the ECG scale. Statistical analysis was performed based on SciPy using Python 3.7 with a designated significance level of 95%.

### Reporting summary

Further information on research design is available in the Nature Portfolio Reporting Summary linked to this article.

## Data availability

The ownership of the raw ECGs belongs to Cardiocloud Medical Technology (Beijing) Co., Ltd. To obtain all raw ECGs, please contact Mr. Wang Jinliang via email at wangjl@cardiocloud.cn, the third contributing author of this paper and head of data at the company. Due to the nature of commercial competition, raw ECGs does not support paid access. It can only be obtained by the demander and the company jointly negotiating and entering into a data sharing agreement or further scientific collaboration agreement. In case of no response from the company within 1 week, the corresponding author of this paper, Mr. Yang Wei with email of weiyanggm@gmail.com, can be contacted to assist in establishing a communication channel. The offline test set total of 7000 ECGs generated in this study for academic purposes have been deposited in the ScienceDB database under accession code https://doi.org/10.57760/sciencedb.07677. Other raw data can be obtained from the corresponding author upon request. The Supplementary Information files accompanying this paper contain all additional relevant data supporting the key findings of this study and can

also be obtained by contacting the corresponding authors upon request. Source data are provided with this paper.

## Code availability

The source code is available at GitHub: https://github.com/SMU-MedicalVision/ECG-Classfication and Zenodo: https://doi.org/10.5281/zenodo.7964774.

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

## Acknowledgements

We are very grateful to the staffs and clinicians at Cardiocloud Medical Technology (Beijing) Co., Ltd. for supporting and annotating the large scale ECG dataset and deploying our model to the server for online testing: team of Yong Yan. This study was funded by the National Key R&D Program of China (no.2018YFC2001203) and the Key Laboratory of Medical Image Processing of Guangdong Provincial (no. 2020B1212060039).

## Author contributions

W.Y. and Q.F. conceived and designed the study, supervised and advanced the project. J.L. designed the experiment, developed the algorithm and wrote the manuscript. H.T. participated in the improvement of the algorithm. J.W. and L.J. collected these ECGs and deployed the model for online test. J.G., B.H. and Y.S. provided medical guidance for our work. All authors reviewed the manuscript.

## Competing interests

The authors declare no competing interests.
