## [Peer Review File · Nature Communications]

Practical intelligent diagnostic algorithm for wearable 12-lead ECG via self-supervised learning on large-scale datasetREVIEWER COMMENTS

Reviewer #1 (Remarks to the Author):

Lai et al. present an important paper regarding automatic analysis of wearable 12-lead ECG. The idea of automatic diagnosis of 12-lead ECG is not novel, but its application on mobile data is new. Few major concerns exist before the paper will suite for publication

1. Mobile devices are used by the public and thus the leads can be replaced. How the automatic system deals with such input? Pleas explain or add explanation in the limitation section
2. Usually more then one disease exists in the 12-lead ECG. How the system deals with that?
3. Please provide reference to the statement in lines 56-57
4. There is a big different if the annotation is performed from scratch (the cardiologist is not aware to the results) or have been reviewed. Please provide statistic of how many diagnoses have been changed by the senior cardiologists.
5. How the annotated and unannotated data divided between the diseases (not just the test)
6. In order to compare the performance of the system to future algorithm performance at least the offline test set must be public

Reviewer #2 (Remarks to the Author):

This is a highly interesting and clinically relevant study by Lai et al where the authors developed an intelligent diagnostic algorithm to detect 55 common cardiac arrhythmias on a wearable 12 lead EKG using self-supervised learning. The authors collected 658,486 wearable 12-lead ECGs, among which 164,538 were diagnosed by a cardiologist and reviewed by senior cardiologists, and the remaining 493,948 ECGs were without diagnosis information. They demonstrate that the model can recognize 55 cardiac events, including normal rhythm, sinus rhythm, eight waveform changes and 45 common arrhythmias, and it achieved the average area under the receiver–operating characteristic curve (AUROC) and average F1 score on offline test of 0.979 and 0.618; the average sensitivity, average specificity and average F1 score during the three-month online test are 0.837, 0.961 and 0.602, respectively. Here are my comments:

- 1: This is a rigorously performed analysis and I commend the authors on their diligence.
- 2: While the average AUROC was near perfect at 0.978, the average AUPRC and F1 score, while significant were only modest at 0.703 and 0.618, respectively, on the offline test set. For the online test set, the average sensitivity, average specificity and average F1 score are 0.837, 0.961 and 0.602, respectively. This is an issue for what the authors propose to be something that would be utilized in a mass scale by the general population. My biggest concern in these results is that it is not sensitive enough. If I understand it correctly, with a sensitivity of 83.7%, it will miss a correct diagnosis in 163 out of 1000 patients. If the authors want to suggest that this tool be applied in a mass scale, then these numbers are suboptimal. Similarly, while specificity is good at 96.1%, it still means 39 out of 1000 patients will have a “false alarm” and a wrong diagnosis. Also, an F1 score of 0.60 is modest. SO how do the authors propose improving these?
- 3: The authors state “Misdiagnosing negative recording as a false positive is nothing more than a cardiologist spending 3 to 5 minutes to diagnose the ECG, and the missed diagnosis of positive data wastes the opportunity for early treatment of patients, with very serious consequences.” This statement does not flow well in the results portion of a high impact manuscript. Please delete it or explain further
- 4: The biggest issue with such technology and the current study is that it may work well in one type of population, but extrapolating into other populations makes the accuracy go down. What happens to results in different skin tone ? Darker skin ? Caucasian skin ? Muscular vs. nonmuscular individuals? Men vs. women? We certainly should have data in multiple ethnicities.
- 5: What do the authors think are the limitations of their study?

Reviewer #3 (Remarks to the Author):

I would like to congratulate the authors for their efforts on this paper. In this study, they use novel machine learning methods to classify >600,000 ECGs collected from a 12 lead wearable ECG monitor into 55 cardiac events. Their novel methods include using multi-scale convolutional networks, addition of pretraining weights and using augmentation operations to clean up artifacts and noise. The diagnostic performance is excellent in general and they externally validated their findings in another Chinese dataset.

I have some major and minor comments

Major comments:

- Regarding the 55 classified cardiac events:

- o What is the difference between normal rhythm and sinus rhythm? In clinical practice, they are the same thing
- o What is a higher p wave? Higher than what? What is the clinical relevance?
- o What is the difference between Q wave abnormal and q wave abnormal? The authors mention they are two subclasses of Q wave changes related to MI. I'm not sure this is true.
- o Did T wave abnormal include peaked T waves?
- o What is the definition of significant sinus bradycardia?
- o PAC nomenclature should be: Conducted PAC, Blocked PAC, and PAC with QRS aberrancy
- o What is paired atrial premature beats? Do you mean bigeminal PACs?
- o What is an atrial escape beat?
- o What is an accelerated atrial rhythm? Do you mean atrial tachycardia?
- o What is the difference between junctional escape and junctional rhythm?
- o Heart enlargement and hypertrophy should just be hypertrophy and this could be left and right ventricular hypertrophy
- o Pacing should be "paced".
- o What is atrial aware paced rhythm? Do you mean atrial sensed, ventricular paced?
- o Should have considered adding biventricular paced rhythm

- Regarding the 55 cardiac events described above, they need to be reviewed again by an expert clinician. Some events are not meaningful clinically, some lack the appropriate clinical nomenclature, and some important ones are missing such as 2nd degree Type 2 heart block, complete heart block. Also, electrolyte abnormalities such as hypokalemia and hyperkalemia would have been good events to assess. Long QT is also important. At least, these are more clinically relevant than sinoatrial block. To make this easy, I suggest the authors remove clinically meaningless and irrelevant diagnoses such as high p wave. They also need to describe how they differentiated things like sinus arrhythmia from sinoatrial exit block.

- The authors present diagnostic performance as averages of AUROC, sensitivity, F1 score, etc... I'm not sure if these variables are normally distributed along the range of the 55 cardiac events. They are probably not and therefore presenting medians and IQRs would be more appropriate.

- Sensitivity of 0.837 is excellent but still maintains a high absolute number of false negatives if the algorithm is extrapolated to millions of patients. I agree that adjusting the classification cutoffs should aim for minimizing false negatives as much as possible rather than false positives. This should be discussed in the limitations section.

- Since most wearable devices on the market are 1-3 lead devices, it would have been interesting to test the diagnostic performance of their algorithms on lead 1 and other limb leads. This would be more relevant to real-world clinical practice. If the authors can do this analysis and add it to the paper, the paper would be stronger. If they can't do the analysis, at least mention in the discussion the fact that developing a similar algorithm for consumer wearables will be a next practical step.

Minor comments:

- What are nonrhythmic arrhythmias? Do the authors mean irregular? Nonrhythmic is a vague clinical

expression.

- Lead misplacement is always an issue with 12 lead ECGs, especially if it is a wearable device and patients are expected to wear it. Did the authors encounter lead displacement in the recordings? If so, what was the rate?
- Please describe further the difference between annotated and unannotated ECGs. What does "annotated according to identity document" mean?
- The article would benefit from adding an example of an ECG with multiple abnormalities. Can the algorithm classify more than one cardiac event in the same ECG? Please describe in the results or methods section how this was dealt with (multiple events on the same ECG)?

Reviewer #4 (Remarks to the Author):

This is a study on the use of AI deep-learning methods to train and deploy an algorithm to automatically recognize 55 ECG diagnoses using a large dataset of wearable 12-lead ECG. The development of an AI-based diagnostic tool for the whole spectrum of ECG abnormalities is still challenging, and the study tries to fill an important gap in the literature. I have some questions and considerations:

1. There is no description of the population from which the ECGs were obtained. What are the mean age, the frequency of men/women, and the pre-existent heart conditions? Why are these patients using a continuous monitoring system? Are they cardiac patients?
2. There is some confusion about the unit used for the development/training, and testing of the neural network algorithm. Indeed, the unnumbered table indicates that from 18778 15-sec 12-lead ECG exams, there are 31686 in normal rhythm, 142486 in sinus rhythm, and 11506 in atrial fibrillation.
3. Please confirm that all 164,538 annotated ECGs were reviewed by three other senior cardiologists. Is this part of the research project, or is it a routine in this service?
4. There is also inconsistency in the way the ECG is classified. Normal rhythm and sinus rhythm are generally synonyms in terms of heart rhythm; Bundle branch block is a general group of abnormalities, not an ECG abnormality itself. ECG diagnoses or classes are not commonly referred to as "events" in cardiology; this term is reserved for cardiovascular events or outcomes, such as myocardial infarction or sudden death.
5. The general accuracy of the algorithm cannot be calculated as the average of the accuracy for each ECG diagnosis. As the authors pointed out, each ECG can have more than one ECG abnormality and the precision should be evaluated for the ECG tracing itself. Please, see a full discussion on this topic at [10.1001/jama.2022.10561](https://doi.org/10.1001/jama.2022.10561).

REVIEWER COMMENTS

Reviewer #1 (Remarks to the Author):

Lai et al. present an important manuscript regarding automatic analysis of wearable 12-lead ECG. The idea of automatic diagnosis of 12-lead ECG is not novel, but its application on mobile data is new.

We are very grateful to Reviewer #1 for the comments and suggestions that helped us to refine our detail of the algorithmic flow and to make the presentation of our work clearer for practical application.

Few major concerns exist before the manuscript will suite for publication:

1. Mobile devices are used by the public and thus the leads can be replaced. How the automatic system deals with such input? Please explain or add explanation in the limitation section.

Response: Lead electrodes can be misplaced during use by non-medical professionals, such as home users. This is known as lead reversal, which occurs when a user first uses a wearable ECG device, and it occurs due to unfamiliarity with the device. For wearable devices, there are six types of lead wearing situations: upper limb left and right lead reversal, upper and lower limb lead reversal, left upper and lower limb lead reversal, right upper and lower limb lead reversal, chest lead reversal, and normal wear connection.

We collected total of 230 lead reversal ECGs in our 164,538 ECG dataset, which were annotated by cardiologists. There is a domain gap between the distribution of lead reversal ECGs and normal wearing ECGs; however, due to the small amount of data, we are currently unable to accurately analyze the impact of this difference on the model classification performance. Our proposed solution is to use a network to determine these six scenarios and provide feedback to the user on their phone within 15 s of the device wearing and signal collection, such that it can be corrected to ensure that all ECGs are collected in normal wearing mode before the data enters the classification model. This is our current project, and the following constitutes a brief description of this part of the work.

We employed the MSDNN as the backbone to train a six-category network to identify the six wearing situations using cross-entropy as a loss function, and deployed it to mobile devices as a initial detection. We recruited 95 volunteers and collected 575 ECGs with a length of 15 s. The training set consisted of 482 ECGs from 80 of these individuals, and the test set consisted of 93 ECGs from the remaining 15 individuals. Of the 93 ECGs tested, only one case of left upper and lower limb lead reversal was misdiagnosed as upper and lower limb lead reversal, the rest of the data were detected correctly. This study has three directions of improvement: recruiting more volunteers to collect richer data, reducing the detection time from 15 s to s ideally 1 s, correcting the wear status instantly when the user starts collecting data, and light-weighting the model for deployment to mobile devices.

Action: We have added "Methods-Pre-filtration" on page 11 of the manuscript as follows:

Pre-filtration of ECGs. *Two factors may affect the quality of ECG signal in wearable ECGs, namely*

lead reversal and artifacts. Lead reversal occurs due to lead electrodes of wearable ECG devices being misplaced on the skin due to unfamiliarity during use by non-medical professionals, such as home users. Artifacts are caused by bad contact or even shedding between the lead electrodes and the skin. These two factors can lead to significant distribution differences between the data collection and normal wearing situations, which severely decreases the classification performance of the model. Our solution is to use the deep learning model to correct the wearing situation of users before uploading ECG, and to exclude the ECG data acquired in an inappropriate wearing situation.

There are six types of device wearing situations: upper limb left and right lead reversal, upper and lower limb lead reversal, left upper and lower limb lead reversal, right upper and lower limb lead reversal, chest lead reversal, and normal wear connection. There are total of 230 lead reversal ECGs in our annotated 164,538 ECGs, all of which are diagnosed by experienced human cardiologists. A domain gap exists between the distribution of lead reversal ECGs and normal wearing ECGs; however, due to the small amount of data, we are currently unable to accurately analyze the impact of this difference on the model classification performance. We used the MSDNN as the backbone to train a six-class network to identify the six wearing situations using cross-entropy as the loss function, and deployed it to mobile devices as a startup detection. We recruited 95 volunteers and collected 575 ECGs with a length of 15 s. The training set consisted of 482 ECGs from 80 of these individuals, and the test set consisted of 93 ECGs from the remaining 15 individuals. Of the 93 data tested, only one case of left upper and lower limb lead reversal was misdiagnosed as upper and lower limb lead reversal, while the rest of the data were detected correctly.

To evaluate artifacts, it is necessary to determine whether the ECG is acceptable via quality assessment. We use the deep learning approach to assess the signal quality of each lead, and filter out ECGs of bad quality that do not have diagnostic value before inputting the data into the cardiac rhythm classification model. This method is described in detail in our previous study [36].

These two works have three directions of future improvement: recruiting more volunteers to collect richer data; reducing the detection time from 15 s to shorter such as ideally 1 s, correcting the wear status instantly when the user starts collecting signal; and light-weighting the model for deployment on mobile devices.

2. Usually more than one disease exists in the 12-lead ECG. How the system deals with that?

Response: Our model currently detects and recognizes 55 cardiac rhythms. A single ECG may have more than one label, in which case it is subjected to the multi-label classification task. The activation function of the network's final layer is sigmoid, and the loss function is a combination of binary cross-entropy and pairwise ranking loss. The model has 55 prediction values between 0 and 1 for each ECG to be diagnosed, and to obtain the final result, 55 thresholds are required to binarize these prediction values, with those greater than or equal to the threshold being 1, and those less than the threshold being 0. The thresholds are different for each cardiac rhythm and are fine-tuned by our senior cardiologists from the default thresholds after taking into account the sensitivity and specificity of each class.

g

Fig. 2-g The operating point selection of class AF on the PR curve.

The graph on the left shows the PR curve for class AF with three operating points marked: Break even point, Optimal F1 point and Fine-tuned point. The graph on the right is the precision, recall and F1 versus threshold from 0 to 1. For the current cardiac rhythm (AF), our default threshold is 0.389 for the break even point, where precision, recall and F1 score are all equal at 0.746. Break even point is not the optimal F1 point, the F1 score peaks at 0.748 corresponding to the threshold of 0.282. However, neither of these two operating points is what we think is most appropriate, and we want the intelligent diagnosis model to be more sensitive and reduce the number of missed positive ECGs (FN), so the threshold was set to 0.1 after fine-tuning by cardiologists.

Threshold	TP	FN	FP	TN	F1	Sensitivity	Specificity
0.000	1314	0	17464	0	0.131	1.000	0.000
0.051	1224	90	1075	16389	0.678	0.932	0.938
0.100	1165	149	767	16697	0.718	0.887	0.956
0.154	1121	193	599	16865	0.739	0.853	0.966
0.205	1078	236	513	16951	0.742	0.820	0.971
0.256	1046	268	450	17014	0.744	0.796	0.974
0.282	1034	280	417	17047	0.748	0.787	0.976
0.308	1015	299	393	17071	0.746	0.772	0.977
0.359	996	318	361	17103	0.746	0.758	0.979
0.389	980	334	334	17130	0.746	0.746	0.981
0.410	964	350	323	17141	0.741	0.734	0.982
0.462	942	372	296	17168	0.738	0.717	0.983
0.513	902	412	266	17198	0.727	0.686	0.985
1.000	0	1314	0	17464	0.000	0.000	1.000

Table 2. Confusion matrix, F1, sensitivity and specificity at different thresholds of atrial fibrillation. Our principle in selecting the operating point is higher sensitivity and fewer false negative samples, but at the same time it will lead to an increase in false positive samples. Even though the false positives of the fine-tuned threshold (767) are double or more than the false positives of the default threshold (334), which will increase the workload of cardiologists, but it is worth it for the access it gives patients.

Action: We have added " Methods-Operating point selection" on page 13 of the manuscript as follows:

Operating point selection. Our model currently detects and recognizes 55 cardiac rhythms. A single ECG may have more than one label and is classified by multi-label classification task. The activation function of network's final layer is sigmoid and the loss function is a combination of

binary cross-entropy and pairwise ranking loss. The model has 55 prediction values ranging between 0 and 1 for each ECG to be diagnosed. To obtain the final result 55 thresholds are required to binarize these prediction values, with those greater than or equal to the threshold being 1 and those less than the threshold being 0. The thresholds vary for each cardiac rhythm and are fine-tuned by our senior cardiologists from the default thresholds after taking into account the sensitivity and specificity of each class.

Fig. 3d and Table 2 show that for the current cardiac rhythm (AF), upon selecting the operating point on the PR curve of offline test set, the threshold of recommended operating point is the break even point of 0.389, where the precision, recall and F1 score are all equal to 0.746. Break even point is not the optimal F1 point, and the F1 score peaks at 0.748 corresponding to a threshold of 0.282. However, both sensitivities at these two operating points are below 0.8, and a lower threshold and higher sensitivity are preferred to reduce the number of missed positives. Nevertheless, this would multiply the number of false positives and increase the workload of cardiologists. Our senior cardiologists finally selected a threshold of 0.1 after systematic consideration, yielding a sensitivity and F1-score of 0.887 and 0.718 respectively. At the same time the false positives of the fine-tuned threshold (767) are double or more than the false positives of the default threshold (334). A single ECG review takes the cardiologist 3-5 minutes generally, but the added expense and cost is well worth it compared to the cost of a missed diagnosis.

3. Please provide reference to the statement in lines 56-57. (Cardiologists can make a diagnosis based on two to three clean lead recordings, but the presence of artifacts in ECGs is a hindrance for computer-intelligent analysis to a certain extent.)

Response: The presence of artifacts in ECG signals may affect the diagnostic results of the model. From a macroscopic perspective, we divided the offline test set into good and inferior quality sets. The division between these two sets is based on the annotation by the human cardiologists. ECGs with excessive artifacts and electrode shedding are of inferior quality. As shown in Figure 2-d, the AUPRC of our model is 0.651 on the offline test set, 0.670 on the high quality set, and 0.611 on the inferior quality set, indicating that the presence of artifacts in ECGs has a negative impact on the diagnostic performance of the model. One of our main contributions of this study is the design of data augmentation to improve the robustness of the model.

From a microscopic perspective, we analyzed the annotation results of two sample ECGs with their corresponding intelligent diagnostic results.

Fig. In this ECG, there are severe artifacts in the first five seconds, and the diagnosis of cardiologists is: normal ECG, sinus rhythm. The diagnosis of the model is normal ECG, sinus rhythm, sinus tachycardia, and right axis deviation, and there is a false alarm occurred.

Fig. There are also serious artifacts in this ECG. The cardiologist's diagnosis was sinus rhythm; T wave abnormal; limited right bundle branch block; and right axis deviation, but the model only diagnosed right axis deviation and there is a missed diagnosis occurred.

Action: At the page 1 we have corrected it to: *Cardiologists can make a diagnosis based on the clear part of an artifact-polluted ECG, however, the presence of artifacts in ECGs poses a challenge*

to computer-intelligent analysis.

4. There is a big different if the annotation is performed from scratch (the cardiologist is not aware to the results) or have been reviewed. Please provide statistic of how many diagnoses have been changed by the senior cardiologists.

Response: Among 164,538 annotated ECGs, annotations of 93,365 ECGs from the two junior and senior cardiologists were identical and passed the review directly, while the annotations of 71,173 ECGs making up 43.26 % of the total were changed.

The annotation process for all ECGs is the same: each ECG is independently annotated and diagnosed by two cardiologists, and then reviewed by two other senior cardiologists who compare inconsistent annotations and submit the final diagnosis. Difficult ECG data that are disputed or for which a definitive diagnosis cannot be given are submitted to an arbitrator for final review and determination.

Action: We have added this to " Results-Dataset " on page 3 of the manuscript as follows:

The gold standard annotation procedure of the ECG dataset is as follows: each ECG was independently diagnosed and annotated by two cardiologists and subsequently reviewed by three other senior cardiologists who compared inconsistent annotations and submitted the final diagnosis. Difficult ECGs that are disputed or for which a definitive diagnosis cannot be given were submitted to an arbitrator for final review and determination. Among the 164,538 annotated ECGs, annotations made on 93,365 ECGs by senior cardiologists and the two cardiologists were identical and passed the review directly, while annotations of the remaining 71,173 ECGs, making up 43.26 % of the total ECGs, were changed. This annotation process is complex and cumbersome, it was conducted only in this research project with the aim to make the model to be more reliable and the evaluation more objective and accurate. In the actual clinical setting, only one cardiologist provides the diagnosis.

5. How the annotated and unannotated data divided between the diseases (not just the test)?

Response: The total number of unannotated ECGs is 493,948 have never been divided. The role of this dataset is to pre-train the Siamese network (MoCo) in conjunction with the training set. The 164,538 annotated ECGs were randomly divided into two sets: the training set with 145,760 ECGs, and the offline test set with 18,778 ECGs. The unannotated ECGs were only used in the pre-training stage, and together with the training set total of 639,708 ECGs to train a Siamese network and save the weights. This weight was then used to initialize the classification network, and the annotated training set was used to train the network, and save the network with the lowest loss function as the final diagnostic model.

The sample size of the unannotated ECGs is approximately four times larger than the annotated ECGs, and the distribution of the unannotated ECGs almost completely covers the distribution of the annotated ECGs, which indicates that fully exploiting the knowledge contained in ECGs itself is expected to improve the classification performance of the diagnostic model. The technological means used is self-supervised learning, which involves two stages: first a contrast learning network is pre-trained using a large amount of ECGs, whose significance is to provide a better initialization

for the downstream classification network than random initialization, in the hope of improving the classification performance of the intelligent diagnostic model. The subsequent stage is a regular classification task that uses the weights from the first stage.

Action: We emphasize in the “Experiment” on page 5 that the unannotated ECGs are used as a whole: *We employed the improved multiscale ResNet18 [20] as the backbone, using a total of 639,708 ECGs of all unannotated ECGs and the training set to pretrain a Siamese convolutional network via momentum contrastive learning, and then used the learned weights as the initialization weights of the downstream classification network. We used the combination of binary cross-entropy and pairwise ranking as the loss function for the multilabel classification task.*

And the annotated ECGs were randomly divided also at page 5: *We have two test sets: the offline dataset and the additional online dataset. Each user was assigned an identity document (ID) in our database, and the same user often uploaded several ECGs. We randomly divided the 164,538 ECGs into a training set with 145,760 samples and the offline test set with 18,778 samples according to ID to ensure that the ECGs of the same individual corresponds to a single dataset.*

6. In order to compare the performance of the system to future algorithm performance at least the offline test set must be public.

Response: The ECG data is owned by Cardiocloud Medical Technology (Beijing) Co., Ltd., which we were allowed to use for scientific research and are obliged to keep confidential, such that neither the data nor the model for this task can be made public. However, if any researchers wish to know the classification performance of their models on our test set, they are invited to send us their models for testing, and we will test and respond as soon as possible.

Furthermore, after carefully considering your suggestion and discussing with the principal of the company, we selected 5,000 ECGs of ST segment changes from all ECGs as a sub-dataset to the one that is publicly available for scientific research. ST change (STC) in the ECG can be classified as ST elevation (STE) and ST depression (STD), which generally indicates the possibility of myocardial infarction and myocardial ischemia in the patient's heart. STE and STD are not mutually exclusive and may occur simultaneously in different leads of the same ECG. The ECG data used were all 12-lead, with a sampling frequency of 500 Hz and a length of 15 s for each ECG signal. data were saved as .mat format files, with label information present in the .mat file and finally summarized in an Excel sheet under the folder. The dataset consisted of 5000 ECGs, of which 4000 ECGs were used for training (containing 180 STEs, 180 STDs, 180 simultaneous STEs and STDs, and 3460 other samples, which were made available to the participants) and 1000 for testing (only for evaluation of the algorithm, which was not made public).

Action: The dataset of “STC ECG Classification” is at <https://www.scidb.cn/anonymous/MOFWTnJx>.

Reviewer #2 (Remarks to the Author):

This is a highly interesting and clinically relevant study by Lai et al where the authors developed an intelligent diagnostic algorithm to detect 55 common cardiac arrhythmias on a

wearable 12 lead EKG using self-supervised learning. The authors collected 658,486 wearable 12-lead ECGs, among which 164,538 were diagnosed by a cardiologist and reviewed by senior cardiologists, and the remaining 493,948 ECGs were without diagnosis information. They demonstrate that the model can recognize 55 cardiac events, including normal rhythm, sinus rhythm, eight waveform changes and 45 common arrhythmias, and it achieved the average area under the receiver–operating characteristic curve (AUROC) and average F1 score on offline test of 0.979 and 0.618; the average sensitivity, average specificity and average F1 score during the three-month online test are 0.837, 0.961 and 0.602, respectively. Here are my comments:

1: This is a rigorously performed analysis and I commend the authors on their diligence.

Response: We are grateful to Reviewer #2 for the careful review of our work, which has helped us to think more deeply about the shortcomings of this work and our future improvements, and has a clearer articulation of the algorithmic process.

2: While the average AUROC was near perfect at 0.978, the average AUPRC and F1 score, while significant were only modest at 0.703 and 0.618, respectively, on the offline test set. For the online test set, the average sensitivity, average specificity and average F1 score are 0.837, 0.961 and 0.602, respectively. This is an issue for what the authors propose to be something that would be utilized in a mass scale by the general population. My biggest concern in these results is that it is not sensitive enough. If I understand it correctly, with a sensitivity of 83.7%, it will miss a correct diagnosis in 163 out of 1000 patients. If the authors want to suggest that this tool be applied in a mass scale, then these numbers are suboptimal. Similarly, while specificity is good at 96.1%, it still means 39 out of 1000 patients will have a “false alarm” and a wrong diagnosis. Also, an F1 score of 0.60 is modest. SO how do the authors propose improving these?

Response: Your understanding of the model's performance is proper, and your comments are pertinent. To answer this question, we must consider three factors carefully:

First, the data are severely unbalanced. In the confusion matrix we listed, the averages of TP, TN, FP, and FN are 533, 9065, 336, and 67, respectively. F1 is calculated as $2TP/(2TP+FP+FN)$, and this indicator is quite sensitive to data imbalance, where the number of FP is only slightly less than TP, which makes F1 appear modest. However, the current results are actually acceptable in terms of sensitivity of 0.837 and specificity of 0.961. We will collect as many positive ECGs as possible in our subsequent work, especially in classes with less than 100 samples, such as ventricular tachycardia (VT).

Second, there is a confrontation between the sensitivity and specificity. Our task is multi-labeled, the activation function of the network's final layer is sigmoid, and the loss function is a combination of binary cross-entropy and pairwise ranking loss. The model has 55 prediction values between 0 and 1 for each ECG to be diagnosed, and to obtain the final result, 55 thresholds are required to binarize these prediction values, with those greater than or equal to the threshold being 1 and those less than the threshold being 0. The thresholds are different for each cardiac rhythm, and they are obtained by our senior cardiologists by fine-tuning the default thresholds after taking into account the sensitivity and specificity of each class.

g

Fig. 2-g The operating point selection of class AF on the PR curve.

The graph on the left shows the PR curve for class AF with three operating points marked: Break even point, Optimal F1 point and Fine-tuned point. The graph on the right is the precision, recall and F1 versus threshold from 0 to 1. For the current cardiac rhythm (AF), our default threshold is 0.389 for the break even point, where precision, recall and F1 score are all equal at 0.746. Break even point is not the optimal F1 point, the F1 score peaks at 0.748 corresponding to the threshold of 0.282. However, neither of these two operating points is what we think is most appropriate, and we want the intelligent diagnosis model to be more sensitive and reduce the number of missed positive ECGs (FN), so the threshold was set to 0.1 after fine-tuning by cardiologists.

For a given class, a threshold of 0 gives a sensitivity of 1 and a precision of 0, and a threshold of 1 gives a sensitivity of 0 and a precision of 1. Our operating point is not the optimal F1 point, and the specificity and precision will decrease while the sensitivity increases. At present, due to the difficulty of the classification task itself, the model cannot be 100% accurate in diagnosing diseases, and there is only a trade-off between sensitivity and specificity that allows the model to make mistakes, and a moderate threshold that is acceptable to human experts to enable the model to perform intelligent diagnostic work.

Third, the ability of current classification models to achieve better performance is mainly limited by two factors: the algorithm level and the difficulty of the task itself. In terms of algorithms, from traditional decision trees, support vector machines, sparse coding, and now mainstream deep learning, the features used in the models are becoming increasingly abundant, and the performance scores on some classification tasks are growing. Currently, we use self-supervised learning, and in future work we will propose specific strategies to improve the classification performance for the wearable ECG classification task. The difficulty of the task itself is currently limited by the small sample size for certain diseases. Ideally, the task itself would be significantly less difficult if the patterns of ECG data in the training set included the patterns of various diseases in the test set. This is why numerous studies are currently collecting more data and employing external data. In future work, we will collect more data to enrich our training set and fundamentally improve the diagnostic performance of the model.

Action: We have added this to "Discussion" on page 10 of the manuscript as follows: *In the three-month online test, the average sensitivity of our model was 0.837, which means that 16.3% of positive ECGs were missed, which could lead to patients missing the precious immediate access to a cardiologist. While the average specificity is good at 0.961, which means that 3.9% of the ECGs*

were false alarms. Falsely alerted ECGs may cause psychological burden to the collector and increase the workload of the cardiologist. The performance of model is limited by the inherent difficulty of the task itself and the power of algorithm and computer, we must weigh the sensitivity and specificity by choose a suitable operating point in the context of allowing the model to make mistakes. Sensitivity and specificity are highly correlated with the choice of the operating point, with sensitivity increasing and specificity decreasing as the threshold decreases. In this regard, we will design specific strategies and methods to the characteristics of the wearable 12-lead ECG and the problems faced in clinical using to improve the classification performance of the model. The average F1 is modest at 0.602, mainly because the optimal F1 point was not selected as the operating point and there is a severe data imbalance in our dataset. The selection of operating points is manually fine-tuned by senior cardiologists, and we will study the automatic threshold selection method. Data imbalance is also a common difficulty in medical data analysis, our current approach to mitigating it is class weighting, and we will use more refined methods to cope with this hindrance in future work.

3: The authors state “Misdiagnosing negative recording as a false positive is nothing more than a cardiologist spending 3 to 5 minutes to diagnose the ECG, and the missed diagnosis of positive data wastes the opportunity for early treatment of patients, with very serious consequences.” This statement does not flow well in the results portion of a high impact manuscript. Please delete it or explain further.

Response: This statement is intended to illustrate a preference in our threshold selection process and is an artificial choice for the second point in the answer to the previous question. When we select threshold operating points on the offline test set, the default threshold represents the break even point or optimal F1 point. However, these are not the best choice, as we would prefer a lower threshold and higher sensitivity to reduce the number of missed positives. Nevertheless, this would multiply the number of false positives and increase the workload of the examiner. However, due to the scarcity of positive samples, the number of false positives is not large in relation to the data volume. A single ECG review takes the physician approximately 3-5 minutes, but the added expense and cost is well worth compared to the cost of a missed diagnosis.

Threshold	TP	FN	FP	TN	F1	Sensitivity	Specificity
0.000	1314	0	17464	0	0.131	1.000	0.000
0.051	1224	90	1075	16389	0.678	0.932	0.938
0.100	1165	149	767	16697	0.718	0.887	0.956
0.154	1121	193	599	16865	0.739	0.853	0.966
0.205	1078	236	513	16951	0.742	0.820	0.971
0.256	1046	268	450	17014	0.744	0.796	0.974
0.282	1034	280	417	17047	0.748	0.787	0.976
0.308	1015	299	393	17071	0.746	0.772	0.977
0.359	996	318	361	17103	0.746	0.758	0.979
0.389	980	334	334	17130	0.746	0.746	0.981
0.410	964	350	323	17141	0.741	0.734	0.982
0.462	942	372	296	17168	0.738	0.717	0.983
0.513	902	412	266	17198	0.727	0.686	0.985

1.000	0	1314	0	17464	0.000	0.000	1.000
-------	---	------	---	-------	-------	-------	-------

Table 2. Confusion matrix, F1, sensitivity and specificity at different thresholds of atrial fibrillation. Our principle in selecting the operating point is higher sensitivity and fewer false negative samples, but at the same time it will lead to an increase in false positive samples. Even though the false positives of the fine-tuned threshold (767) are double or more than the false positives of the default threshold (334), which will increase the workload of cardiologists, but it is worth it for the access it gives patients.

Action: We have explained in "Results-Diagnostic performance" on page 7 of the manuscript as follows: *The diagnosis of ECG is essentially the screening of possible cardiac diseases, and we are committed to detecting as many arrhythmic ECGs from the huge amount of data as possible to be diagnosed for further diagnosis by cardiologists. Although misdiagnosing negative recording as a false positive increases the time cost and workload of the cardiologist, a missed diagnosis of positive data wastes the opportunity for early treatment of patients, with very serious consequences. For each cardiac rhythm, a threshold is required for the model to define whether the intelligent diagnostic result is positive. As shown in Fig. 2e, take the atrial fibrillation for example, the common recommended operating point on PR curve is the break even point or the optimal F1 point, but our senior cardiologists chose the fine-tuned point after considering recall and precision, where the recall is superior and the precision and F1-score are also acceptable. The F1 scores for both test sets are modest due to the large number of false positives generated by the severe data imbalance, but this is actually acceptable. As in the table 2, even if the number of false positive cases is two, three or more times the number of true positive cases, this order of magnitude is insignificant compared to the large number of negative cases, and the specificity of 0.961 indicates that a large number of true negative ECGs are not misdiagnosed. In terms of sensitivity, the value of 0.837 indicates that most of the positive samples were detected. Therefore, our model can actually reduce the burden on cardiologists and has a great role and potential for real-time diagnosis and remote diagnosis.*

4: The biggest issue with such technology and the current study is that it may work well in one type of population, but extrapolating into other populations makes the accuracy go down. What happens to results in different skin tone? Darker skin? Caucasian skin? Muscular vs. nonmuscular individuals? Men vs. women? We certainly should have data in multiple ethnicities.

Response: The utility of deep learning models is currently limited by the geography, age, population, and devices. The fundamental solution to bridging this domain gap is to collect sufficient ECGs from a diverse range of individuals. Our current data sources are all in China and cover all provinces, autonomous regions, and municipalities. A total of 67.3 % of the data is collected from male participants, and the remaining 32.8 % are female. The age range was from 8 to 94 years. Because the abnormality of cardiac rhythms was not correlated with factors such as age and gender, we did not make additional considerations and designs for these factors. The main application of our work in the future is aimed at Chinese cardiac patients. If there is a need to expand the model to be applicable to different devices, different skin colors, and sports populations in the future, the preferred option would be to collect sufficient ECG data from these populations, and the second-best option would be to collect a certain amount of data and implement a domain adaptation strategy

based on the existing model. We aim to collect data from multiple races and populations to promote our work in the future. We propose a feasible solution for the automated diagnostic application of wearable ECG that could provide a reference for researchers involved in this field.

Action: We have added this to " Results-Dataset " on page 3 of the manuscript as follows: *Our ECGs were collected across China and cover all provinces, autonomous regions, and municipalities. Three main sources of these data included home users wearing the wearable devices to collect ECGs at home, patients in hospitals who use wearable devices, and companies and the hospital organizing free ECG diagnostic events to collect ECG data. A total of 67.3% of the dataset was comprised from male participants, and the remaining 32.8 % were female. The ages of these 76,482 individuals ranged from 8 to 94 years, with an average age of 51 years. Because the anomalies in cardiac rhythms were not significantly associated with age or sex, we did not make additional considerations and designs for these factors. Of the total 164,538 ECGs, approximately 31686 samples constituted healthy normal ECGs, with the remaining 132,852 ECGs exhibiting varying arrhythmia.*

5: What do the authors think are the limitations of their study?

Response: Thanks to your thoughts and suggestions, we have reorganized our manuscript and considered feasible directions and methods to improve our work in the future.

In terms of limitations of this application, there are two points:

1. Our model is currently only applicable to the Chinese population using a specific wearable ECG device. In future work, we will expand our data sources to collect ECG data from different ethnic groups, populations, and devices, and attempt to bridge the domain gap between some of the data distributions to extend this work to a wider range of applications.
2. There are symbiotic and mutually exclusive relationships between multiple fine-grained labels, and we must apply this a priori knowledge to the training and testing of the model, to obtain more reasonable and natural diagnostic results with better classification performance.

In terms of limitations of the algorithm, there are three relevant points:

1. The data augmentation is not sufficient, and the artificially simulated artifacts are consistently inferior to the real ones. We aim to investigate the decoupling of signal and artifacts in ECG data with interference in the future, and decompose the artifacts present in the wearable ECG signal as a novel data augmentation scheme to further improve the classification performance of the model.
2. For the large amount of unannotated ECGs, only self-supervised pre-training method was employed. In future work, we will attempt to combine this with semi-supervised methods to further mine the knowledge contained in the ECG signal itself and improve the generalization performance of the model.
3. The solution for dealing with data imbalances currently uses only label weighting, we will attempt more detailed solutions to mitigate the effects of this phenomenon, reduce overfitting of the model, and produce fewer false positives or false negatives in prediction.

Action: We further elaborated the "Discussion" on page 11 of the manuscript as follows: *At the algorithm level, even if we simulate several kinds of interference that wearable ECGs may encounter in certain scenes, our efforts are far from enough due to more complicated predicaments in the real environment encountered by wearable ECG devices. During the online test, cardiologist feedback*

told us that our model was mainly missing and misdiagnosing the ECG data that were contaminated by interference, and we will study the decoupling of signal and artifacts in ECG with interference in the future, and coupling the ECG with artifacts decomposed from other ECG as a new data augmentation operation to further improve the classification performance of the model. In addition, the improvement of classification performance via self-supervised learning is not significant, and we will combine semi-supervised learning methods to further mine the knowledge contained in the ECG signal itself and improve the generalization performance of the model. Finally, there are symbiotic and mutually exclusive relationships between multiple fine-grained labels, and we should apply this a priori knowledge to the method to obtain more reasonable and natural diagnostic results.

Although deep learning has been successfully applied to physiological signal processing, our model can be initially applied to clinical diagnostic settings. However, our model is currently only applicable to the population using the specific wearable ECG devices within China. In future work, we will expand our data sources to collect ECGs from different ethnic groups, populations and devices, and try to bridge the domain gap between the distribution of ECGs from different sources, try our best to recognize more cardiac rhythms at a fine-grained level and extend this work to a wider range of applications. In conclusion, we have established a baseline as far as the practicability of deep learning approach, which can provide a reference and feasible report for other researchers. The difficulties encountered by the model in real-world settings are more complex and there is much more work we can study, which provides more opportunities and challenges for the analysis and processing of ECG.

Reviewer #3 (Remarks to the Author):

I would like to congratulate the authors for their efforts on this manuscript. In this study, they use novel machine learning methods to classify >600,000 ECGs collected from a 12 lead wearable ECG monitor into 55 cardiac events. Their novel methods include using multi-scale convolutional networks, addition of pretraining weights and using augmentation operations to clean up artifacts and noise. The diagnostic performance is excellent in general and they externally validated their findings in another Chinese dataset.

I have some major and minor comments.

We are very grateful to reviewer #3 for pointing out the shortcomings in our work and helping us to refine the revisions in the algorithm flow, data set, definition and translation of cardiac rhythms.

Major comments:

- Regarding the 55 classified cardiac events:

o What is the difference between normal rhythm and sinus rhythm? In clinical practice, they are the same thing.

Response: Normal rhythm is a mistake in translation and must be translated as normal ECG, which is a holistic conclusion of the current ECG. The sinus rhythm is what is commonly recognized as normal rhythm. For the ECG of a healthy individual, the typical diagnosis is: normal ECG, sinus

rhythm.

Action: Normal rhythm is corrected to normal ECG on page 3.

o What is a higher p wave? Higher than what? What is the clinical relevance?

Response: We consulted the literature and concluded that higher p wave is more appropriately translated as a high or tall P wave, defined as high or tall P waves in leads II, III, and AVF, with the amplitude of the P wave being greater than 0.25 mV in limb leads, and greater than 0.20 mV in chest leads. If low voltage exists, the amplitude of the P wave must be greater than half of the amplitude of R wave in the same lead. Clinically, this generally indicates right atrial enlargement.

Action: Higher p wave is corrected to high p wave on page 3.

o What is the difference between Q wave abnormal and q wave abnormal? The authors mention they are two subclasses of Q wave changes related to MI. I'm not sure this is true.

Response: Normal q-wave duration is less than 0.04 s, and the depth is less than 1/4 of the same lead R-wave amplitude. If both the time duration is greater than 0.04 s, and the depth is greater than 1/4 R amplitude, it is referred to as the Q wave abnormal. If only one of these two conditions is met, it is referred to as the q wave abnormal. Any abnormality of the q-wave in lead III is the Q wave abnormal.

Action: This definition is added in Supplementary Table 1.

o Did T wave abnormal include peaked T waves?

Response: These 55 labels are a subset of the total number of labels listed by cardiologists in the database, and the total number of pre-defined labels is 294. These classes cannot all be made public due to the property right, and only those that have been tested online and are listed in the manuscript can be detailed. These labels have two levels in the catalogue, namely the parent catalogue with a certain number of secondary catalogues.

Parent catalogue	Secondary catalogue	remark
normal ECG	normal ECG	holistic conclusion of ECG
sinus rhythm	sinus rhythm sinus tachycardia sinus bradycardia significant sinus bradycardia sinus arrhythmia ...	normal rhythm greater than 100 beats per minute less than 60 beats per minute less than 45 beats per minute sinus arrest, sinus premature beats, etc.
heart enlargement and hypertrophy	high left ventricular voltage high right ventricular voltage widened P wave high P wave	generally indicates left atrial enlargement generally indicates right atrial enlargement
	Q wave abnormal	normal q: time duration<0.04s, depth<1/4 R

myocardial infarction	q wave abnormal poor R wave progression	Q wave abnormal: time duration>0.04s and depth>1/4 R q wave abnormal: time duration>0.04s or depth>1/4 R
ST change	ST elevation ST depression	
T wave abnormal	peaked T wave flattening T wave T-wave inversion biphasic T waves T wave notching	
premature atrial contraction	atrial premature beats atrial premature beat not transmitted atrial premature beats with aberrations paired atrial premature beats ...	atrial bigeminy, atrial trigeminy, etc.
atrial escape beat	atrial escape beat accelerated atrial escape beat slow atrial escape beat	three or more consecutive atrial escape beats in an ECG is atrial rhythm
atrial escape rhythm	atrial escape rhythm accelerated atrial escape rhythm slow atrial escape rhythm	
atrial fibrillation	atrial fibrillation rapid atrial fibrillation AF with intraventricular aberrant conduction ...	paroxysmal AF, etc.
atrial flutter	atrial flutter	
junctional escape beat	junctional escape beat accelerated junctional escape beat slow junctional escape beat	three or more consecutive junctional escape beats in an ECG is junctional rhythm
junctional rhythm	junctional rhythm accelerated junctional rhythm slow junctional rhythm	
supraventricular tachycardia	atrial tachycardia ...	junctional tachycardia, etc.
premature ventricular contractions	ventricular premature beats ventricular bigeminy ...	ventricular trigeminy, etc.
ventricular tachycardia	ventricular tachycardia ...	burst/multifocal ventricular tachycardia, etc.
pre-excitation syndrome	pre-excitation syndrome	
axis and voltages	left axis deviation right axis deviation indeterminate cardiac axis low limb leads voltages low chest leads voltages clockwise rotation early repolarization ...	counterclockwise rotation, etc.
atrioventricular block	1st degree atrioventricular block 2nd degree atrioventricular block-I 2nd degree atrioventricular block-II ...	3rd degree atrioventricular block, etc.
bundle branch block	left anterior fascicular block complete right bundle branch block incomplete right bundle branch block	

	complete left bundle branch block ...	left posterior fascicular block, etc.
2nd sinoatrial block	2nd sinoatrial block	
paced rhythm	atrial paced rhythm ventricular paced rhythm atrial-ventricular paced rhythm atrial sensed ventricular paced rhythm ...	atrial/ventricular paced fusion wave, etc.

Supplementary Table 1: The subordination of the 55 labels (marked in red) that our model can currently recognize. We have a total of 294 labels in our ECG database, and these labels have two levels of catalogue. Our ultimate goal is to use a deep learning model to recognize and detect all these labels, but the lack of samples is the main factor limiting the classification performance of the model to minority classes. Therefore, when the amount of samples in the secondary catalogue is insufficient, we can only merge some of the secondary labels and then recognize their parent catalogue. In the future, we will make our best efforts to collect ECGs of minority class with a small number of samples, and to expand the range of recognizable labels as much as possible so that more cardiac rhythms can be recognized by the model.

The T wave abnormal is likewise a parent catalogue that includes T wave abnormal traces on any lead such as the peaked T wave, flattening T wave, T-wave inversion, biphasic T waves, and T wave notching.

Action: This subordination is added in Supplementary Table 1.

o What is the definition of significant sinus bradycardia?

Response: Significant sinus bradycardia is defined as a sinus rhythm of less than 45 beats per minute.

Action: This definition is added in Supplementary Table 1.

o PAC nomenclature should be: Conducted PAC, Blocked PAC, and PAC with QRS aberrancy

Response: PAC is also a parent catalogue that includes atrial premature beat, atrial premature beat not transmitted, atrial premature beats with aberrations, paired atrial premature beats, atrial bigeminy and atrial trigemini.

Action: This subordination is added in Supplementary Table 1.

o What is paired atrial premature beats? Do you mean bigeminal PACs?

Response: Paired atrial premature beats are not bigeminal PACs or atrial bigeminy. Paired atrial premature beats are defined as two consecutive atrial premature beats; bigeminal PACs or atrial bigeminy are defined as one atrial premature beat immediately following each sinus rhythm beat.

Action: This definition is added in Supplementary Table 1.

o What is an atrial escape beat?

Response: Atrial escape beat also is atrial escape, which is defined as an ectopic beat generated by a passive impulse from other potential pacing sites in the atria when there is a delay in the emission of excitation or the presence of efferent block in the sinus node or atria, causing the heart to excite. Three or more such events occur in succession, creating an atrial escape rhythm.

Action: This definition is added in Supplementary Table 1.

o What is an accelerated atrial rhythm? Do you mean atrial tachycardia?

Response: This is a translation error, it should be corrected to accelerated atrial escape rhythm, which is defined as the presence of three or more consecutive atrial escape beats in a segment of the ECG, also known as nonparoxysmal atrial tachycardia. Atrial tachycardia is already in the current list of recognizable cardiac rhythms.

Action: This definition is added in Supplementary Table 1.

o What is the difference between junctional escape and junctional rhythm?

Response: Junctional rhythm is defined as the presence of three or more consecutive junctional escape beats in a segment of the ECG.

Action: This definition is added in Supplementary Table 1.

o Heart enlargement and hypertrophy should just be hypertrophy and this could be left and right ventricular hypertrophy

Response: Heart enlargement and hypertrophy are also a parent catalogue that includes high left ventricular voltage, high right ventricular voltage, high P wave and widened P wave.

Action: This subordination is added in Supplementary Table 1.

o Pacing should be “paced”.

Response and Action: Thank you for correcting our translation, all the “pacing” has been corrected to “paced” on page 3.

o What is atrial aware paced rhythm? Do you mean atrial sensed, ventricular paced?

Response and Action: This is a translation error, and the “atrial aware ventricular paced rhythm” is corrected to “atrial sensed ventricular paced rhythm” on page 3.

o Should have considered adding biventricular paced rhythm

Response: The number of labels in our database is 294, many of which only have few samples,

such as ventricular tachycardia (VT), whose sample size is less than 100. Currently, the lack of samples is the main factor limiting the classification performance of the model to rare classes. In the future, we will make our best efforts to collect ECGs with a small number of samples, and to expand the range of recognizable labels as much as possible such that more cardiac rhythms can be recognized by the model.

Action: This section was added to the “Discussion” on page 11 of the manuscript: *Although deep learning has been successfully applied to physiological signal processing, our model can be initially applied to clinical diagnostic settings. However, our model is currently only applicable to the population using the specific wearable ECG devices within China. In addition, the number of classes in our database is 294, many of which only have few samples such as ventricular tachycardia (VT) whose sample size is less than 100. Currently, the lack of samples is the main factor limiting the classification performance of the model to rare classes. In future work, we will expand our data sources to collect ECGs from different ethnic groups, populations and devices, and try to bridge the domain gap between the distribution of ECGs from different sources, try our best to recognize more cardiac rhythms at a fine-grained level and extend this work to a wider range of applications. In conclusion, we have established a baseline as far as the practicability of deep learning approach, which can provide a reference and feasible report for other researchers. The difficulties encountered by the model in real-world settings are more complex and there is much more work we can study, which provides more opportunities and challenges for the analysis and processing of ECG.*

- Regarding the 55 cardiac events described above, they need to be reviewed again by an expert clinician. Some events are not meaningful clinically, some lack the appropriate clinical nomenclature, and some important ones are missing such as 2nd degree Type 2 heart block, complete heart block. Also, electrolyte abnormalities such as hypokalemia and hyperkalemia would have been good events to assess. Long QT is also important. At least, these are more clinically relevant than sinoatrial block. To make this easy, I suggest the authors remove clinically meaningless and irrelevant diagnoses such as high p wave. They also need to describe how they differentiated things like sinus arrhythmia from sinoatrial exit block.

Response: Thank you for your review from the perspective of a clinical expert in cardiology. Our cardiac rhythms or labels were designed with reference to the AAMI (Association for the Advancement of Medical Instrumentation) standards, the “AHA/ACCF/HRS recommendations for the standardization and interpretation of the electrocardiogram (2007)”, and the “Chinese expert consensus statement on the standardization of electrocardiography diagnostic terminology 2019”. We have reviewed the translation of all labels, and organized and added the label relationships and definitions to the supplementary information, and hope that more readers can understand our work more from the clinical relevance and make suggestions.

Action: We carefully verified the translation of the various cardiac rhythms on page 3 of the manuscript and added Supplementary Table 1.

- The authors present diagnostic performance as averages of AUROC, sensitivity, F1 score,

etc... I'm not sure if these variables are normally distributed along the range of the 55 cardiac events. They are probably not and therefore presenting medians and IQRs would be more appropriate.

Response: It is true that these labels are not normally distributed, and there are symbiosis and mutual exclusion relationships between some cardiac rhythms. We have carefully considered your suggestion and use violin plots to replace partial graphs in Fig. 2, which shows the IQRs of the evaluation indicator. In addition, the simple average basically reflects the classification performance of the model, gives an intuitive impression when briefly describing our work, and facilitates direct comparison with related work of other researchers, so the average method in the table 1 is retained. In the future, we will investigate richer and more objective evaluation systems to assess the performance of our ECG intelligent diagnosis.

Action: We use violin plots to replace partial graphs in Fig. 2 on page 4 of the manuscript:

- Sensitivity of 0.837 is excellent but still maintains a high absolute number of false negatives if the algorithm is extrapolated to millions of patients. I agree that adjusting the classification cutoffs should aim for minimizing false negatives as much as possible rather than false positives. This should be discussed in the limitations section.

Response: Thank you for agreeing with our preferences and views. When we select threshold operating points on the offline test set, we would prefer a lower threshold and a higher sensitivity

to reduce the number of missed positives, but this would multiply the number of false positives and increase the workload of the examiner. However, due to the scarcity of positive samples, the number of false positives is not large in relation to the volume of data. A single ECG review can take the physician 3-5 minutes generally, but the added expense and cost is well worth it compared to the cost of a missed diagnosis.

Action: We have explained in "Results-Diagnostic performance" on page 7 of the manuscript as follows: *The diagnosis of ECG is essentially the screening of possible cardiac diseases, and we are committed to detecting as many arrhythmic ECGs from the huge amount of data as possible to be diagnosed for further diagnosis by cardiologists. Although misdiagnosing negative recording as a false positive increases the time cost and workload of the cardiologist, a missed diagnosis of positive data wastes the opportunity for early treatment of patients, with very serious consequences. For each cardiac rhythm, a threshold is required for the model to define whether the intelligent diagnostic result is positive. As shown in Fig. 2e, take the atrial fibrillation for example, the common recommended operating point on PR curve is the break even point or the optimal F1 point, but our senior cardiologists chose the fine-tuned point after considering recall and precision, where the recall is superior and the precision and F1-score are also acceptable. The F1 scores for both test sets are modest due to the large number of false positives generated by the severe data imbalance, but this is actually acceptable. As in the table 2, even if the number of false positive cases is two, three or more times the number of true positive cases, this order of magnitude is insignificant compared to the large number of negative cases, and the specificity of 0.961 indicates that a large number of true negative ECGs are not misdiagnosed. In terms of sensitivity, the value of 0.837 indicates that most of the positive samples were detected. Therefore, our model can actually reduce the burden on cardiologists and has a great role and potential for real-time diagnosis and remote diagnosis.*

- Since most wearable devices on the market are 1-3 lead devices, it would have been interesting to test the diagnostic performance of their algorithms on lead 1 and other limb leads. This would be more relevant to real-world clinical practice. If the authors can do this analysis and add it to the manuscript, the manuscript would be stronger. If they can't do the analysis, at least mention in the discussion the fact that developing a similar algorithm for consumer wearables will be a next practical step.

Response: Most of the widely used on the market wearable ECG devices are 1-3 leads, while our object of study is 12-lead wearable ECG signals. Electrodes of different leads are placed at different locations on the skin surface of the human body, which can be considered as observing the electrophysiological activity of the heart from multiple perspectives. Theoretically, the higher the number of leads, the richer the information observed, and the more accurate the diagnostic results. We used 1-3 of the 12 leads to simulate the signals acquired from 1-3 wearable ECG devices respectively: 1. Using lead I to simulate the single-lead device; 2. Using II, V1 and V5 to simulate a 3-lead Holter ECG device used in the hospital 3. Using I, avf, and V2 to simulate a 3-lead Frank orthogonal lead ECG device. It should be noted that in these three simulation scenarios really 1-3 leads are only slightly similar to the listed sub-leads in waveforms, but they are not part of any of the 12 leads. We compared the classification performances of these three cases with the 12-lead

ECG, and the average AUPRC are 0.490, 0.666, 0.640 and 0.703 respectively. It can be seen that the single-lead signal is much less informative than the 3-lead and 12-lead, and the 3-lead ECG also not as informative as the 12-lead ECG.

Fig. 2-e Classification performance when only partial leads are used.

We currently do not have mature ECGs and annotation on 1-3 leads for study, and the 1-3 lead ECG signal cannot simply be regarded as a sub of the 12 leads, and there is a huge difference in analysis between the simulative leads and the true 1-3 leads. However, from the above results, the amount of information in the 12-lead wearable ECG is certainly richer than that in the 1-3 lead ECG, and our work has more practical advantages. We think we can put this part of the analysis in the discussion section: the easiest and simplest solution to extend our work based on 12-lead ECG to 1-3-lead ECG is to study the generation from 1-3 lead signals to 12-lead signals, converting 1-3 lead ECGs acquired by other devices to the 12-lead ECGs we currently use, which can be directly diagnosed using our trained model. Getting the level of 12-lead ECG device diagnostic results with a cheaper and more convenient device can be a direction of our future work.

Action: We have added "Extending the model based on 12-lead ECG to 1-3 lead ECG devices" on page 9 of the manuscript as follows:

Extending the model based on 12-lead ECG to 1-3 lead ECG devices. Most of the widely used wearable ECG devices on the market are 1-3 leads, while the object of our study is 12-lead wearable ECG signals. In order to extend the applied range of our model to these 1-3 lead ECG devices, we propose a feasible solution: using data generation methods such as generative adversarial networks (GAN) to transform 1-3 leads ECG into the currently used 12 leads ECG, which then can be straightforward to use our model. However, we currently do not have mature 1-3 leads ECGs dataset and corresponding annotation for study.

To explore possible benefits of such behavior, we used 1-3 of the 12 leads to simulate the signals acquired from 1-3 wearable ECG devices: using lead I to simulate the single-lead device; using II, VI and V5 to simulate the 3-lead Holter ECG device used in the hospital; using I, AVF and V2 to simulate the 3-lead Frank orthogonal lead ECG device. Notably, in these three simulation scenarios really 1-3 leads are only similar to the listed sub-leads in waveforms, albeit they are not part of any

of the 12 leads. We compared the classification performances of these three cases with the 12-lead ECG as Fig. 2e, yielding the average AUPRC are 0.490, 0.666, 0.640 and 0.703, respectively. The single-lead signal is significantly less informative than the 3-lead and 12-lead, and the 3-lead ECG also not as informative as the 12-lead ECG. We consider that the higher the number of leads would yield to the observation of richer information and yield more accurate the diagnostic results. This approach obtains the intelligent diagnostic results on the level of the 12-lead ECG device with a lower cost and more convenient device.

Minor comments:

- What are nonrhythmic arrhythmias? Do the authors mean irregular? Nonrhythmic is a vague clinical expression.

Response: Our intention is to illustrate that the application scenario of wearable ECGs is broader than conventional ECGs in hospitals. When a patient goes to hospital and records an ECG, the acquisition time is often less than 1 minute, and the rhythm that appears in this ECG is often rhythmic, meaning that the change in cardiac rhythm occurs in each heartbeat. The non-rhythmicity that can be detected by a wearable ECG over a long period of time is mainly transient, occurring only once or several times over a long period of time, and is not reflected in each heartbeat. The definitions of rhythmicity and non-rhythmicity are indeed rather vague, and our formulation is lacking.

Action: We have take your suggestions into account to tighten the wording and amend the relevant parts of the manuscript.

- Lead misplacement is always an issue with 12 lead ECGs, especially if it is a wearable device and patients are expected to wear it. Did the authors encounter lead displacement in the recordings? If so, what was the rate?

Response: Lead electrodes can be misplaced during use by non-medical professionals, such as home users. This is known as lead reversal, which occurs when a user first uses a wearable ECG device, and it occurs due to unfamiliarity with the device. For wearable devices, there are six types of lead wearing situations: upper limb left and right lead reversal, upper and lower limb lead reversal, left upper and lower limb lead reversal, right upper and lower limb lead reversal, chest lead reversal, and normal wear connection.

We collected total of 230 lead reversal ECGs in our 164,538 ECG dataset, which were annotated by cardiologists. There is a domain gap between the distribution of lead reversal ECGs and normal wearing ECGs; however, due to the small amount of data, we are currently unable to accurately analyze the impact of this difference on the model classification performance. Our proposed solution is to use a network to determine these six scenarios and provide feedback to the user on their phone within 15 s of the device wearing and signal collection, such that it can be corrected to ensure that all ECGs are collected in normal wearing mode before the data enters the classification model. This is our current project, and the following constitutes a brief description of this part of the work.

We employed the MSDNN as the backbone to train a six-category network to identify the six

wearing situations using cross-entropy as a loss function, and deployed it to mobile devices as a initial detection. We recruited 95 volunteers and collected 575 ECGs with a length of 15 s. The training set consisted of 482 ECGs from 80 of these individuals, and the test set consisted of 93 ECGs from the remaining 15 individuals. Of the 93 ECGs tested, only one case of left upper and lower limb lead reversal was misdiagnosed as upper and lower limb lead reversal, the rest of the data were detected correctly. This study has three directions of improvement: recruiting more volunteers to collect richer data, reducing the detection time from 15 s to ideally 1 s, correcting the wear status instantly when the user starts collecting data, and light-weighting the model for deployment to mobile devices.

Action: We have added "Methods-Pre-filtration" on page 11 of the manuscript as follows:

Pre-filtration of ECGs. *Two factors may affect the quality of ECG signal in wearable ECGs, namely lead reversal and artifacts. Lead reversal occurs due to lead electrodes of wearable ECG devices being misplaced on the skin due to unfamiliarity during use by non-medical professionals, such as home users. Artifacts are caused by bad contact or even shedding between the lead electrodes and the skin. These two factors can lead to significant distribution differences between the data collection and normal wearing situations, which severely decreases the classification performance of the model. Our solution is to use the deep learning model to correct the wearing situation of users before uploading ECG, and to exclude the ECG data acquired in an inappropriate wearing situation.*

There are six types of device wearing situations: upper limb left and right lead reversal, upper and lower limb lead reversal, left upper and lower limb lead reversal, right upper and lower limb lead reversal, chest lead reversal, and normal wear connection. There are total of 230 lead reversal ECGs in our annotated 164,538 ECGs, all of which are diagnosed by experienced human cardiologists. A domain gap exists between the distribution of lead reversal ECGs and normal wearing ECGs; however, due to the small amount of data, we are currently unable to accurately analyze the impact of this difference on the model classification performance. We used the MSDNN as the backbone to train a six-class network to identify the six wearing situations using cross-entropy as the loss function, and deployed it to mobile devices as a startup detection. We recruited 95 volunteers and collected 575 ECGs with a length of 15 s. The training set consisted of 482 ECGs from 80 of these individuals, and the test set consisted of 93 ECGs from the remaining 15 individuals. Of the 93 data tested, only one case of left upper and lower limb lead reversal was misdiagnosed as upper and lower limb lead reversal, while the rest of the data were detected correctly.

To evaluate artifacts, it is necessary to determine whether the ECG is acceptable via quality assessment. We use the deep learning approach to assess the signal quality of each lead, and filter out ECGs of bad quality that do not have diagnostic value before inputting the data into the cardiac rhythm classification model. This method is described in detail in our previous study [36].

These two works have three directions of future improvement: recruiting more volunteers to collect richer data; reducing the detection time from 15 s to shorter such as ideally 1 s, correcting the wear status instantly when the user starts collecting signal; and light-weighting the model for deployment on mobile devices.

- Please describe further the difference between annotated and unannotated ECGs. What does “annotated according to identity document” mean?

Response: This statement “The offline test set comprised 18,778 randomly selected annotated ECGs according to the identity document, and the remainder of the ECGs constitute the training set.” illustrates how to divide the training set and the offline test set. Each user is given an ID number in our database, and the same user will often upload numerous ECGs. If the ECGs from the same individual are divided into both the training set and the offline test set, the apparent performance of the model will be greater than the real performance due to information leakage. Therefore, we divide the training set and offline test set by user ID, and ECGs from the same ID in the offline test set will never appear in the training set, making the evaluation results more objective.

Action: Our presentation may be ambiguous, and we have updated the presentation of this section in the last paragraph of the “Results-Dataset” on page 5 in the manuscript: *We have two test sets: the offline dataset and the additional online dataset. Each user was assigned an identity document (ID) in our database, and the same user often uploaded several ECGs. We randomly divided the 164,538 ECGs into a training set with 145,760 samples and the offline test set with 18,778 samples according to ID to ensure that the ECGs of the same individual corresponds to a single dataset. In addition, we deployed the network to the server and provided real-time artificial intelligence diagnosis for 10,000 ECGs uploaded by users who use the same wearable ECG device from February to May 2022. These recordings were also diagnosed by the cardiologist committee and all used for online testing. These ECGs conform to the distribution of patients in the real world, and positive samples of some classes are very rare, which allows more objective evaluation of the detection and classification performance of our approach.*

- The article would benefit from adding an example of an ECG with multiple abnormalities. Can the algorithm classify more than one cardiac event in the same ECG? Please describe in the results or methods section how this was dealt with (multiple events on the same ECG)?

Response: Our model currently detects and recognizes 55 cardiac rhythms. A single ECG may have more than one label, in which case it is subjected to the multi-label classification task. The activation function of the network’s final layer is sigmoid, and the loss function is a combination of binary cross-entropy and pairwise ranking loss. The model has 55 prediction values between 0 and 1 for each ECG to be diagnosed, and to obtain the final result, 55 thresholds are required to binarize these prediction values, with those greater than or equal to the threshold being 1, and those less than the threshold being 0. The thresholds are different for each cardiac rhythm and are fine-tuned by our senior cardiologists from the default thresholds after taking into account the sensitivity and specificity of each class.

g

Fig. 2-g The operating point selection of class AF on the PR curve.

The graph on the left shows the PR curve for class AF with three operating points marked: Break even point, Optimal F1 point and Fine-tuned point. The graph on the right is the precision, recall and F1 versus threshold from 0 to 1. For the current cardiac rhythm (AF), our default threshold is 0.389 for the break even point, where precision, recall and F1 score are all equal at 0.746. Break even point is not the optimal F1 point, the F1 score peaks at 0.748 corresponding to the threshold of 0.282. However, neither of these two operating points is what we think is most appropriate, and we want the intelligent diagnosis model to be more sensitive and reduce the number of missed positive ECGs (FN), so the threshold was set to 0.1 after fine-tuning by cardiologists.

Threshold	TP	FN	FP	TN	F1	Sensitivity	Specificity
0.000	1314	0	17464	0	0.131	1.000	0.000
0.051	1224	90	1075	16389	0.678	0.932	0.938
0.100	1165	149	767	16697	0.718	0.887	0.956
0.154	1121	193	599	16865	0.739	0.853	0.966
0.205	1078	236	513	16951	0.742	0.820	0.971
0.256	1046	268	450	17014	0.744	0.796	0.974
0.282	1034	280	417	17047	0.748	0.787	0.976
0.308	1015	299	393	17071	0.746	0.772	0.977
0.359	996	318	361	17103	0.746	0.758	0.979
0.389	980	334	334	17130	0.746	0.746	0.981
0.410	964	350	323	17141	0.741	0.734	0.982
0.462	942	372	296	17168	0.738	0.717	0.983
0.513	902	412	266	17198	0.727	0.686	0.985
1.000	0	1314	0	17464	0.000	0.000	1.000

Table 2. Confusion matrix, F1, sensitivity and specificity at different thresholds of atrial fibrillation. Our principle in selecting the operating point is higher sensitivity and fewer false negative samples, but at the same time it will lead to an increase in false positive samples. Even though the false positives of the fine-tuned threshold (767) are double or more than the false positives of the default threshold (334), which will increase the workload of cardiologists, but it is worth it for the access it gives patients.

Action: We have added " Methods-Operating point selection" on page 13 of the manuscript as follows:

Operating point selection. Our model currently detects and recognizes 55 cardiac rhythms. A single ECG may have more than one label and is classified by multi-label classification task. The activation function of network's final layer is sigmoid and the loss function is a combination of

binary cross-entropy and pairwise ranking loss. The model has 55 prediction values ranging between 0 and 1 for each ECG to be diagnosed. To obtain the final result 55 thresholds are required to binarize these prediction values, with those greater than or equal to the threshold being 1 and those less than the threshold being 0. The thresholds vary for each cardiac rhythm and are fine-tuned by our senior cardiologists from the default thresholds after taking into account the sensitivity and specificity of each class.

Fig. 3d and Table 2 show that for the current cardiac rhythm (AF), upon selecting the operating point on the PR curve of offline test set, the threshold of recommended operating point is the break even point of 0.389, where the precision, recall and F1 score are all equal to 0.746. Break even point is not the optimal F1 point, and the F1 score peaks at 0.748 corresponding to a threshold of 0.282. However, both sensitivities at these two operating points are below 0.8, and a lower threshold and higher sensitivity are preferred to reduce the number of missed positives. Nevertheless, this would multiply the number of false positives and increase the workload of cardiologists. Our senior cardiologists finally selected a threshold of 0.1 after systematic consideration, yielding a sensitivity and F1-score of 0.887 and 0.718 respectively. At the same time the false positives of the fine-tuned threshold (767) are double or more than the false positives of the default threshold (334). A single ECG review takes the cardiologist 3-5 minutes generally, but the added expense and cost is well worth it compared to the cost of a missed diagnosis.

Reviewer #4 (Remarks to the Author):

This is a study on the use of AI deep-learning methods to train and deploy an algorithm to automatically recognize 55 ECG diagnoses using a large dataset of wearable 12-lead ECG. The development of an AI-based diagnostic tool for the whole spectrum of ECG abnormalities is still challenging, and the study tries to fill an important gap in the literature. I have some questions and considerations:

We are grateful to reviewer #4 for reviewing our work, pointing out our gaps in a professional medical orientation, and helping us improve the interpretation and description of methodological details, data sets, and disease names.

1. There is no description of the population from which the ECGs were obtained. What are the mean age, the frequency of men/women, and the pre-existent heart conditions? Why are these patients using a continuous monitoring system? Are they cardiac patients?

Response: Our current data are all collected across China and cover all provinces, autonomous regions, and municipalities. A total of 67.3 % of the data is collected from male participants, and the remaining 32.8 % are female. The age range was from 8 to 94 years, with an average age of 51 years. As the abnormality of the cardiac rhythms does not correlate with age or gender, we did not make additional considerations and designs for these factors. The main subjects of future application of our work are Chinese cardiac patients. Of the total 164,538 ECGs, approximately 31686 samples were healthy normal ECGs, with the remaining 132,852 ECGs exhibiting varying arrhythmia. There are three main sources of these data: users who wear these wearable devices to collect ECGs at home, patients in hospitals who use wearable devices, and the company and the hospital organize

free ECG diagnostic events to collect ECG data. The majority of these ECG contributors are cardiac patients, along with several healthy people as the control group.

Action: We have updated the “Results-Dataset” on page 3 of the manuscript to include a description of the dataset collection population as follows: *Our ECGs were collected across China and cover all provinces, autonomous regions, and municipalities. Three main sources of these data included home users wearing the wearable devices to collect ECGs at home, patients in hospitals who use wearable devices, and companies and the hospital organizing free ECG diagnostic events to collect ECG data. A total of 67.3% of the dataset was comprised from male participants, and the remaining 32.8 % were female. The ages of these 76,482 individuals ranged from 8 to 94 years, with an average age of 51 years. Because the anomalies in cardiac rhythms were not significantly associated with age or sex, we did not make additional considerations and designs for these factors. Of the total 164,538 ECGs, approximately 31686 samples constituted healthy normal ECGs, with the remaining 132,852 ECGs exhibiting varying arrhythmia.*

2. There is some confusion about the unit used for the development/training, and testing of the neural network algorithm. Indeed, the unnumbered table indicates that from 18778 15-sec 12-lead ECG exams, there are 31686 in normal rhythm, 142486 in sinus rhythm, and 11506 in atrial fibrillation.

Response: Thank you very much for pointing out our error, it was indeed an oversight on our part to inadvertently combine the number of samples from different rhythm categories into the offline test set when collating the excel sheet.

Action: Table 1 on page 6 of the manuscript has been corrected as follows:

Cardiac rhythms	Samples (n=164538)	Offline test set (n=18778)			Online test set (n=10000)						
		AUROC	AUPRC	F1	TP	TN	FP	FN	Sensitivity	Specificity	F1
NECG	31686	0.946	0.760	0.720	1786	7187	700	327	0.845	0.911	0.777
SR	142486	0.981	0.995	0.976	8478	1142	212	168	0.981	0.843	0.978
HP	3463	0.985	0.760	0.669	107	9772	108	13	0.892	0.989	0.639
QAb	12437	0.971	0.796	0.702	522	9109	243	126	0.806	0.974	0.739
qAb	6705	0.945	0.479	0.517	180	9338	384	98	0.647	0.961	0.428
PRP	11947	0.963	0.699	0.649	444	8880	542	134	0.768	0.942	0.568
STE	7548	0.948	0.621	0.512	516	8737	589	158	0.766	0.937	0.580
STD	27266	0.937	0.775	0.613	1301	7302	1237	160	0.890	0.855	0.651
TAbs	49719	0.967	0.923	0.857	2372	6534	785	309	0.885	0.893	0.813
PT	1878	0.970	0.705	0.630	353	9430	138	79	0.817	0.986	0.765
ST	8086	0.988	0.896	0.812	294	9536	138	32	0.902	0.986	0.776
SB	18247	0.983	0.886	0.835	925	8531	453	91	0.910	0.950	0.773
SSB	405	0.989	0.518	0.506	40	9876	68	16	0.714	0.993	0.488
SA	16394	0.933	0.712	0.651	515	8504	771	210	0.710	0.917	0.512
PAC	10042	0.981	0.904	0.844	545	9134	190	131	0.806	0.980	0.773
APNT	946	0.970	0.472	0.501	90	9560	320	30	0.750	0.968	0.340
APA	1959	0.991	0.802	0.696	74	9711	187	28	0.725	0.981	0.408
PAP	936	0.980	0.371	0.403	85	9639	244	32	0.726	0.975	0.381
AEB	1126	0.977	0.595	0.527	240	9359	320	81	0.748	0.967	0.545
AAER	623	0.989	0.695	0.623	96	9636	261	7	0.932	0.974	0.417
AF	8463	0.982	0.827	0.718	369	9493	76	62	0.856	0.992	0.842
RAF	1941	0.989	0.605	0.582	88	9843	52	17	0.838	0.995	0.718
AFIVBC	1103	0.989	0.655	0.527	13	9950	33	4	0.765	0.997	0.413
AFL	3961	0.976	0.671	0.615	107	9827	37	29	0.787	0.996	0.764
JE	873	0.990	0.609	0.429	77	9528	372	23	0.770	0.962	0.281

JR	360	0.980	0.248	0.186	112	9757	113	18	0.862	0.989	0.631
SVT	3225	0.969	0.692	0.494	306	9420	243	31	0.908	0.975	0.691
AT	2197	0.960	0.564	0.551	98	9607	257	38	0.721	0.974	0.399
PVC	8240	0.975	0.919	0.781	966	8874	74	86	0.918	0.992	0.924
VB	221	0.998	0.428	0.513	246	9655	49	50	0.831	0.995	0.832
VT	91	0.997	0.710	0.593	26	9939	15	20	0.565	0.998	0.598
PES	593	0.988	0.817	0.812	54	9848	88	10	0.844	0.991	0.524
LAD	4606	0.984	0.693	0.652	320	8959	676	45	0.877	0.930	0.470
RAD	15444	0.949	0.643	0.648	1016	8445	296	243	0.807	0.966	0.790
ICA	1149	0.996	0.753	0.641	130	9577	286	7	0.949	0.971	0.470
LLLV	841	0.995	0.700	0.426	140	9299	544	17	0.892	0.945	0.333
LCLV	1474	0.992	0.555	0.517	150	9533	296	21	0.877	0.970	0.486
CR	1727	0.981	0.454	0.360	108	9489	376	27	0.800	0.962	0.349
ER	2925	0.961	0.423	0.279	272	8889	776	63	0.812	0.920	0.393
IABV	11633	0.979	0.771	0.683	491	8549	912	48	0.911	0.904	0.506
IIAVBI	321	0.997	0.858	0.788	61	9787	135	17	0.782	0.986	0.445
BBB	26093	0.972	0.892	0.820	1648	7444	707	201	0.891	0.913	0.784
LAFB	1319	0.988	0.679	0.512	140	9688	147	25	0.848	0.985	0.619
CRBBB	4844	0.998	0.965	0.947	383	9557	50	10	0.975	0.995	0.927
IRBBB	11294	0.973	0.715	0.654	917	8364	585	134	0.873	0.935	0.718
CLBBB	1224	0.999	0.965	0.896	204	9701	75	20	0.911	0.992	0.811
IISAB	266	0.978	0.295	0.263	61	9823	74	42	0.592	0.993	0.513
HEH	16865	0.948	0.805	0.623	604	7479	1860	57	0.914	0.801	0.387
HLVV	10231	0.986	0.903	0.786	376	9097	496	31	0.924	0.948	0.588
HRVV	421	0.977	0.544	0.473	33	9905	50	12	0.733	0.995	0.516
PR	2170	0.994	0.938	0.912	434	9389	163	14	0.969	0.983	0.831
APR	435	0.992	0.625	0.372	67	9836	95	2	0.971	0.990	0.580
VPR	572	0.999	0.875	0.764	70	9642	285	3	0.959	0.971	0.327
AVPR	475	0.998	0.773	0.188	130	9670	194	6	0.956	0.980	0.565
ASVPR	386	0.992	0.723	0.704	111	9799	79	11	0.910	0.992	0.712
Average	-	0.978	0.703	0.618	533	9065	336	67	0.837	0.961	0.602

3. Please confirm that all 164,538 annotated ECGs were reviewed by three other senior cardiologists. Is this part of the research project, or is it a routine in this service?

Response: The annotated ECGs (164,538+10,000) were divided into three parts: a training set, an offline test set, and an online test set with sample sizes of 145,760, 18,778, and 10,000 respectively. The training and offline test set were collected from 2016 to 2022, and the online test set was collected from February to May 2022. The annotation process for all ECGs is the same: each ECG is independently annotated and diagnosed by two cardiologists and then reviewed by three other senior cardiologists who compare the inconsistent annotations and submit the final diagnosis. Difficult ECG data that are disputed or for which a definitive diagnosis cannot be given are submitted to an arbitrator for final review and determination. This annotation process is complex and cumbersome, and in addition to the annotation there is also review, which serves to train the model to be more reliable and the test results to be more objective and accurate. In the actual clinical setting of ECG diagnosis, only one cardiologist provides the diagnosis.

Action: We have updated the “Results-Dataset” on page 3 of the manuscript of the annotation procedure as follows: *The gold standard annotation procedure of the ECG dataset is as follows: each ECG was independently diagnosed and annotated by two cardiologists and subsequently reviewed by three other senior cardiologists who compared inconsistent annotations and submitted the final diagnosis. Difficult ECGs that are disputed or for which a definitive diagnosis cannot be given were submitted to an arbitrator for final review and determination. Among the 164,538*

annotated ECGs, annotations made on 93,365 ECGs by senior cardiologists and the two cardiologists were identical and passed the review directly, while annotations of the remaining 71,173 ECGs, making up 43.26 % of the total ECGs, were changed. This annotation process is complex and cumbersome, it was conducted only in this research project with the aim to make the model to be more reliable and the evaluation more objective and accurate. In the actual clinical setting, only one cardiologist provides the diagnosis.

4. There is also inconsistency in the way the ECG is classified. Normal rhythm and sinus rhythm are generally synonyms in terms of heart rhythm; Bundle branch block is a general group of abnormalities, not an ECG abnormality itself. ECG diagnoses or classes are not commonly referred to as "events" in cardiology; this term is reserved for cardiovascular events or outcomes, such as myocardial infarction or sudden death.

Response: Normal rhythm is a mistake in translation and must be translated as normal ECG, which is a holistic conclusion of the current ECG. The sinus rhythm is what is commonly recognized as normal rhythm. For the ECG of a healthy individual, the typical diagnosis is: normal ECG, sinus rhythm.

Action: Normal rhythm is corrected to normal ECG on page 3.

Response: Our model detects and recognizes 55 cardiac rhythms that have a two-level catalogue, with some labels, such as the bundle branch block being the parent catalogue with 11 secondary catalogues: the left anterior fascicular block, left posterior fascicular block, complete left bundle branch block, intermittent complete left bundle branch block, incomplete left bundle branch block, intermittent incomplete left bundle branch block, complete right bundle branch block, intermittent complete right bundle branch block, incomplete right bundle branch block, intermittent incomplete right bundle branch block, and limited right bundle branch block. Currently, only four secondary catalogues are in the diagnosable labels, and the remaining secondary catalogues can only be merged into the parent catalogue to yield an approximate result, because the corresponding sample size is too small. In future work, we will collect a larger dataset ECGs to produce more fine-grained diagnostic results.

Action: This subordination is added in Supplementary Table 1.

Parent catalogue	Secondary catalogue	remark
normal ECG	normal ECG	holistic conclusion of ECG
sinus rhythm	sinus rhythm	normal rhythm
	sinus tachycardia	greater than 100 beats per minute
	sinus bradycardia	less than 60 beats per minute
	significant sinus bradycardia	less than 45 beats per minute
	sinus arrhythmia	
	...	sinus arrest, sinus premature beats, etc.
heart enlargement and hypertrophy	high left ventricular voltage	
	high right ventricular voltage	
	widened P wave	generally indicates left atrial enlargement
	high P wave	generally indicates right atrial enlargement
myocardial infarction	Q wave abnormal	normal q: time duration<0.04s, depth<1/4 R
	q wave abnormal	Q wave abnormal: time duration>0.04s and depth>1/4 R
	poor R wave progression	q wave abnormal: time duration>0.04s or depth>1/4 R

ST change	ST elevation ST depression	
T wave abnormal	peaked T wave flattening T wave T-wave inversion biphasic T waves T wave notching	
premature atrial contraction	atrial premature beats atrial premature beat not transmitted atrial premature beats with aberrations paired atrial premature beats ...	atrial bigeminy, atrial trigeminy, etc.
atrial escape beat	atrial escape beat accelerated atrial escape beat slow atrial escape beat	three or more consecutive atrial escape beats in an ECG is atrial rhythm
atrial escape rhythm	atrial escape rhythm accelerated atrial escape rhythm slow atrial escape escape rhythm	
atrial fibrillation	atrial fibrillation rapid atrial fibrillation AF with intraventricular aberrant conduction ...	paroxysmal AF, etc.
atrial flutter	atrial flutter	
junctional escape beat	junctional escape beat accelerated junctional escape beat slow junctional escape beat	three or more consecutive junctional escape beats in an ECG is junctional rhythm
junctional rhythm	junctional rhythm accelerated junctional rhythm slow junctional rhythm	
supraventricular tachycardia	atrial tachycardia ...	junctional tachycardia, etc.
premature ventricular contractions	ventricular premature beats ventricular bigeminy ...	ventricular trigeminy, etc.
ventricular tachycardia	ventricular tachycardia ...	burst/multifocal ventricular tachycardia, etc.
pre-excitation syndrome	pre-excitation syndrome	
axis and voltages	left axis deviation right axis deviation indeterminate cardiac axis low limb leads voltages low chest leads voltages clockwise rotation early repolarization ...	counterclockwise rotation, etc.
atrioventricular block	1st degree atrioventricular block 2nd degree atrioventricular block-I 2nd degree atrioventricular block-II ...	3rd degree atrioventricular block, etc.
bundle branch block	left anterior fascicular block complete right bundle branch block incomplete right bundle branch block complete left bundle branch block ...	left posterior fascicular block, etc.

2nd sinoatrial block	2nd sinoatrial block	
paced rhythm	atrial paced rhythm ventricular paced rhythm atrial-ventricular paced rhythm atrial sensed ventricular paced rhythm ...	atrial/ventricular paced fusion wave, etc.

Supplementary Table 1: The subordination of the 55 labels (marked in red) that our model can currently recognize. We have a total of 294 labels in our ECG database, and these labels have two levels of catalogue. Our ultimate goal is to use a deep learning model to recognize and detect all these labels, but the lack of samples is the main factor limiting the classification performance of the model to minority classes. Therefore, when the amount of samples in the secondary catalogue is insufficient, we can only merge some of the secondary labels and then recognize their parent catalogue. In the future, we will make our best efforts to collect ECGs of minority class with a small number of samples, and to expand the range of recognizable labels as much as possible so that more cardiac rhythms can be recognized by the model.

Response: These 55 labels contain the normal ECG, heartbeat waveform change, and various arrhythmias. The use of “events” as a generic term is indeed slightly inappropriate, and we decided to change the generic term of these labels from “cardiac events” to “cardiac rhythms”.

Action: We changed the title of the manuscript to: Practical cardiac rhythms diagnostic algorithm for wearable 12-lead ECG via self-supervised learning on large-scale dataset.

5. The general accuracy of the algorithm cannot be calculated as the average of the accuracy for each ECG diagnosis. As the authors pointed out, each ECG can have more than one ECG abnormality and the precision should be evaluated for the ECG tracing itself. Please, see a full discussion on this topic at 10.1001/jama.2022.10561.

Response: Indeed, we count the statistics of these indicators using a simple average rather than a weighted average of these classes, and we do not take into account the respective costs or losses of false positives and negatives for each class of intelligent diagnostic results. We also wanted to use more appropriate indicators to accurately assess this work and considered the cost-sensitive loss strategy. To this end, we invited a panel of cardiologists to manually assign different cost weights to each cardiac rhythm. However, this study failed, and its weights are difficult to design reasonably when the number of labels is larger than 10. Moreover, due to the large number of labels and complex hierarchical relationships, we currently have 55 practical labels, and more fine-grained intelligent diagnosis will be achieved in the future as the number of samples from other labels increases and as technology advances. Therefore, we have not yet completed the manual design of cost or weight of each cardiac rhythm due to the subjective nature and difficulty posed by complex relationships.

Currently, this overall simple average reflects the classification performance of the model, yields an intuitive impression when briefly describing our work, and facilitates direct comparison with related work of other researchers, such that we use direct instead of weighted averaging. In the future, we aim to investigate richer and more objective evaluation systems to assess the performance of our ECG intelligent diagnosis.

REVIEWER COMMENTS

Reviewer #2 (Remarks to the Author):

I have no further comments. While the authors have addressed my comments appropriately, there are some fundamental shortcomings which they cannot correct. The biggest one is its generalizability in a broader population with different skin tones and muscle mass etc.

Reviewer #4 (Remarks to the Author):

Thank you for the comprehensive review of the article considering the reviewers' comments. About my suggestions, I would like to stress the following:

1. The authors added a satisfactory description of the population from which the ECGs were obtained.
2. Numbers in Tables 1 and 6 of the manuscript were corrected.
3. They detailed the process of review of the ECGs by cardiologists.
4. They added Supplementary Table 1 with a hierarchical list of ECG labels that answer some of the inconsistencies of the original classification.

*** However, the authors maintain using the wrong term "cardiac rhythms" for all ECG diagnostic classes. It should be clear that only when the ECG abnormality reflects an arrhythmic disorder someone can say that this is a "cardiac rhythm" class. Many of the "labels" used by the authors are standard morphological ECG abnormalities (Q wave abnormalities, high LV voltage, ST elevation). So, it is completely wrong to say that these abnormalities are "cardiac Rhythm" classes. This must be changed in a final version.

*** There are inconsistencies and unacceptable absences in the list of the Supplementary Table 1. So there is no complete AV block, and the third column does not match very well with the 2nd one.

5. I wrote: "The general accuracy of the algorithm cannot be calculated as the average of the accuracy for each ECG diagnosis. As the authors pointed out, each ECG can have more than one ECG

abnormality and the precision should be evaluated for the ECG tracing itself." The authors did not provide the general accuracy of the method considering the right diagnosis per ECG, not per ECG class. Please, provide it.

REVIEWER COMMENTS

1. Please disclose the classification labels that you claim to be under property rights. We would be unable to consider your manuscript further in absence of these.

Response: Thank you for your advice, and we have listed all the ECG labels in Supplementary Table 1 of the Supplementary Information.

Parent catalogue	Secondary catalogue	Remark
normal ECG	normal ECG	holistic conclusion of ECG
sinus rhythm	sinus rhythm sinus tachycardia sinus bradycardia significant sinus bradycardia sinus arrhythmia wandering atrial pacemaker wandering pacemaker within sinoatrial node sinus arrest long RR interval wandering pacemaker within atrioventricular node sinus premature beats sinus bigeminy	approach to normal rhythm greater than 100 beats per minute less than 60 beats per minute less than 45 beats per minute
premature atrial contraction	atrial premature beats atrial bigeminy frequent atrial premature beats atrial premature beat not transmitted atrial premature beats with aberrations paired atrial premature beats atrial trigeminy atrial reciprocal beats multifocal atrial premature beats atrial fusion wave	
supraventricular premature beats	junctional premature beats paired junctional premature beats junctional bigeminy frequent junctional premature beats junctional fusion wave junctional trigeminy atrioventricular node reentry	
atrial escape beat	atrial escape beat accelerated atrial escape beat slow atrial escape beat	three or more consecutive atrial escape beats in an ECG is atrial rhythm
atrial escape rhythm	atrial escape rhythm accelerated atrial escape rhythm slow atrial escape escape rhythm	
atrial fibrillation	atrial fibrillation atrial fibrillation with rapid ventricular rate atrial fibrillation with slow ventricular rate AF with intraventricular aberrant conduction paroxysmal atrial fibrillation	

	impure atrial fibrillation short paroxysmal atrial fibrillation linking phenomenon	
atrial flutter	atrial flutter short paroxysmal atrial flutter impure atrial flutter atrial flutter with 2:1 conduction atrial flutter with variable conduction	
junctional escape beat	junctional escape beat accelerated junctional escape beat slow junctional escape beat	three or more consecutive junctional escape beats in an ECG is junctional rhythm
junctional rhythm	junctional rhythm accelerated junctional rhythm slow junctional rhythm	
interfering atrioventricular dissociation	complete interfering atrioventricular dissociation incomplete interfering atrioventricular dissociation	
supraventricular tachycardia	paroxysmal supraventricular tachycardia atrial tachycardia junctional tachycardia short paroxysmal atrial tachycardia multifocal atrial tachycardia atrial tachycardia with variable conduction supraventricular tachycardia with intraventricular aberrant conduction	
premature ventricular contractions	ventricular premature beats paired ventricular premature beats ventricular bigeminy frequent ventricular premature beats ventricular trigeminy multifocal ventricular premature beats R on T premature ventricular complex ventricular reciprocal beats interpolated PVC	
ventricular arrhythmia	ventricular chaotic ventricular chaotic with intraventricular aberrant conduction escape capture bigeminy escape capture trigeminy ventricular parasystole ventricular fusion beat ventricular peristaltic wave accelerated idioventricular rhythm accelerated multifocal idioventricular rhythm ventricular flutter ventricular fibrillation	
arrest	asystole sudden cardiac arrest	
ventricular escape beat	ventricular escape beat accelerated ventricular escape beat slow ventricular escape beat	
ventricular escape rhythm	ventricular escape rhythm	

	slow ventricular escape rhythm	
ventricular tachycardia	ventricular tachycardia burst ventricular tachycardia paroxysmal ventricular tachycardia multifocal ventricular tachycardia TDP ventricular tachycardia bundle branch reentrant ventricular tachycardia polymorphic ventricular tachycardia bidirectional ventricular tachycardia wide QRS complex tachycardia	
pre-excitation syndrome	ventricular preexcitation intermittent ventricular preexcitation	
short PR interval	short PR interval	
axis and voltages	left axis deviation right axis deviation indeterminate cardiac axis low limb leads voltages low chest leads voltages low voltage on the 12 leads counterclockwise rotation clockwise rotation dextrocardia early repolarization paced AV delay epsilon wave J wave P-QRS-T alternans P wave alternans QRS complex alternans SISISIII syndrome T wave alternans R/S wave ratio greater than 1 in lead V1	
atrioventricular block	1st degree atrioventricular block 2nd degree atrioventricular block-I 2nd degree atrioventricular block-II high-grade atrioventricular block 2:1 atrioventricular block 3rd degree atrioventricular block intermittent 1st degree atrioventricular block atrioventricular dissociation	
2nd sinoatrial block	2nd sinoatrial block 2nd degree sinoatrial block-I 2nd degree sinoatrial block-II	
bundle branch block	left anterior fascicular block left posterior fascicular block complete left bundle branch block intermittent complete left bundle branch block incomplete left bundle branch block intermittent incomplete left bundle branch block	

	complete right bundle branch block intermittent complete right bundle branch block incomplete right bundle branch block intermittent incomplete right bundle branch block partial right bundle branch block nonspecific intraventricular conduction disturbance	
other blocks	intraventricular block intra-atrial block	
heart enlargement and hypertrophy	high left ventricular voltage high right ventricular voltage widened P wave high P wave	generally indicates left atrial enlargement generally indicates right atrial enlargement
paced rhythm	atrial paced rhythm ventricular paced rhythm atrial-ventricular paced rhythm atrial sensed ventricular paced rhythm	
pacemaker malfunction	pacemaker undersensing pacemaker oversensing pacemaker dysfunction	
ST elevation	ST elevation in lead I ST elevation in lead II ST elevation in lead III ST elevation in lead AVR ST elevation in lead AVL ST elevation in lead AVF ST elevation in lead V1 ST elevation in lead V2 ST elevation in lead V3 ST elevation in lead V4 ST elevation in lead V5 ST elevation in lead V6	ST segment change (including ST elevation and ST depression)
ST depression	ST depression in lead I ST depression in lead II ST depression in lead III ST depression in lead AVR ST depression in lead AVL ST depression in lead AVF ST depression in lead V1 ST depression in lead V2 ST depression in lead V3 ST depression in lead V4 ST depression in lead V5 ST depression in lead V6	
ST segment duration	prolongation of ST segment duration ST Segment shortening	
T-wave changes	T-wave changes in lead I T-wave changes in lead II T-wave changes in lead III T-wave changes in lead AVR	T wave abnormal (including T-wave changes, T-wave inversion, biphasic T waves and peaked T wave)

	T-wave changes in lead AVL T-wave changes in lead AVF T-wave changes in lead V1 T-wave changes in lead V2 T-wave changes in lead V3 T-wave changes in lead V4 T-wave changes in lead V5 T-wave changes in lead V6	
T-wave inversion	T-wave inversion in lead I T-wave inversion in lead II T-wave inversion in lead III T-wave inversion in lead AVR T-wave inversion in lead AVL T-wave inversion in lead AVF T-wave inversion in lead V1 T-wave inversion in lead V2 T-wave inversion in lead V3 T-wave inversion in lead V4 T-wave inversion in lead V5 T-wave inversion in lead V6	
biphasic T waves	biphasic T waves in lead I biphasic T waves in lead II biphasic T waves in lead III biphasic T waves in lead AVR biphasic T waves in lead AVL biphasic T waves in lead AVF biphasic T waves in lead V1 biphasic T waves in lead V2 biphasic T waves in lead V3 biphasic T waves in lead V4 biphasic T waves in lead V5 biphasic T waves in lead V6	
peaked T wave	peaked T wave in lead I peaked T wave in lead II peaked T wave in lead III peaked T wave in lead AVR peaked T wave in lead AVL peaked T wave in lead AVF peaked T wave in lead V1 peaked T wave in lead V2 peaked T wave in lead V3 peaked T wave in lead V4 peaked T wave in lead V5 peaked T wave in lead V6	
elevated u wave	elevated u wave	
Q-Tc interval	Q-Tc interval prolongation shortening Q-Tc interval	
Brugada syndrome	Brugada syndrome	
Q wave abnormal	Q wave abnormal in lead I	

	Q wave abnormal in lead II Q wave abnormal in lead III Q wave abnormal in lead AVR Q wave abnormal in lead AVL Q wave abnormal in lead AVF Q wave abnormal in lead V1 Q wave abnormal in lead V2 Q wave abnormal in lead V3 Q wave abnormal in lead V4 Q wave abnormal in lead V5 Q wave abnormal in lead V6	normal q: time duration<0.04s, depth<1/4 R Q wave abnormal: time duration>0.04s and depth>1/4 R q wave abnormal: time duration>0.04s or depth>1/4 R
q wave abnormal	q wave abnormal in lead I q wave abnormal in lead II q wave abnormal in lead III q wave abnormal in lead AVR q wave abnormal in lead AVL q wave abnormal in lead AVF q wave abnormal in lead V1 q wave abnormal in lead V2 q wave abnormal in lead V3 q wave abnormal in lead V4 q wave abnormal in lead V5 q wave abnormal in lead V6	
poor R wave progression	poor R wave progression in lead V1 poor R wave progression in lead V2 poor R wave progression in lead V3 poor R wave progression in lead V4 poor R wave progression in lead V5 poor R wave progression in lead V6	

Supplementary Table 1: All ECG labels and its subordination of our database. These labels are two levels of catalogue, and the 60 labels that our model can currently recognize are marked in red. Our ultimate goal is to use a deep learning model to recognize and detect all these labels, but the lack of rare class samples is the main factor limiting the classification performance of the model to practical. Therefore, when the amount of samples in the secondary catalogue is insufficient, we can only merge some of the secondary labels and then recognize their parent catalogue. In the future, we will make our best efforts to collect ECGs of rare class with a small number of samples, and to expand the range of recognizable labels as much as possible so that more cardiac rhythms can be recognized by the model.

2. The reviewer finds that the following comment: "Some events are not meaningful clinically, some lack the appropriate clinical nomenclature, and some important ones are missing such as 2nd degree Type 2 heart block, complete heart block. Also, electrolyte abnormalities such as hypokalemia and hyperkalemia would have been good events to assess. Long QT is also important. At least, these are more clinically relevant than sinoatrial block. To make this easy, I suggest the authors remove clinically meaningless and irrelevant diagnoses such as high p wave. They also need to describe how they differentiated things like sinus arrhythmia from sinoatrial exit block." was left unanswered. Please provide a response to this comment that fully addresses the related concern.

Response: After careful thought and discussion, we have removed the following 5 labels that are not meaningful or relevant in the clinical setting: left axis deviation, right axis deviation, indeterminate cardiac axis, clockwise rotation and high P wave. At the same time, the following 10 more important labels were added: elevated U wave, Brugada Syndrome, atrial bigeminy, paired ventricular premature beats, ventricular arrhythmia, ventricular fusion beat, ventricular escape beat, 3rd degree atrioventricular block, limited right bundle branch block and nonspecific intraventricular conduction disturbance. However, there are some particularly important labels, such as Q-Tc interval

prolongation and 2nd degree atrioventricular block-II, which cannot be applied in a clinical setting at present because their sample size is too small or even less than 50 and the AUPRC is inferior at less than 0.1. Our model therefore ended up with a total of 60 labels for which a diagnosis could be made, and we updated all experiments in the manuscript.

Electrolyte abnormalities such as hyperkalaemia and hypokalaemia are mainly diagnosed by the blood test for potassium. There are non-specific changes in the ECG in the presence of hyperkalemia or hypokalemia, and no label corresponds exactly to hyperkalaemia and hypokalaemia. For hyperkalaemia, the ECG changes are: peaked T waves, shortening of the QT interval, widening of the QRS, ST depression, disappearance of the P-wave, sinus ventricular conduction, and when the potassium level increases above 10.0, the QRS wave may fuse with the T-wave in a sinusoidal curve, resulting in sinus arrest and ventricular tachycardia. In general, the ECG in hypokalaemia shows elevated U waves, low flat T waves, T-U fusion, prolonged Q-T, premature beats and various tachyarrhythmias. We can only use the model to detect as many ECG abnormalities as possible so that cardiologists can make a comprehensive diagnosis taking into account the patient's symptoms and other test results.

Only second degree sinoatrial block is marked in our database and can be classified as second degree type I sinoatrial block and second degree type II sinoatrial block. Second degree type 1 sinoatrial block also named Wenckebach block, means there is a delay in the conduction from the sinoatrial node to the atrium and this delay increases gradually until one impulse is completely blocked and a loss of P-wave occurs. The P-P interval is gradually decreased. The ensuing pause is twice as long as the cardiac cycle preceding the block. The P-P interval after the pause is longer than the P-P interval before the pause. In second degree type II sinoatrial block impulses are blocked sporadically, and the pause between the visible beats are always multiples of the normal P-P interval. Usually there will be 2 to 4 P-P intervals between the beats (implying that one, two or three sinus impulses are blocked). Sinus arrhythmias are mainly manifested by sinus P waves; the difference in P-P interval is greater than 0.12s at the same time.

Action: We redid all the experiments in the manuscript after updating the labels and reran the online test between January and February 2023, and we state that the new experimental results are consistent with the conclusions obtained from the previous results. Below is a table of the test performance of our model, which we have updated on page 6 of the manuscript.

Rhythm classes	Samples (n=164538)	Offline test set (7000)			Online test set (12521)						
		AUROC	AUPRC	$F1_{label}$	TP_{label}	TN_{label}	FP_{label}	FN_{label}	Sen_{label}	Spe_{label}	$F1_{label}$
NECG	31686	0.952	0.753	0.649	2134	9111	764	512	0.807	0.923	0.770
SR	142486	0.984	0.994	0.965	9200	2046	310	965	0.905	0.868	0.935
QAb	12437	0.953	0.746	0.774	347	11607	434	133	0.723	0.964	0.550
qAb	6705	0.935	0.509	0.528	51	11990	405	75	0.405	0.967	0.175
PRP	11947	0.906	0.510	0.460	446	11139	842	94	0.826	0.930	0.488
STE	7548	0.937	0.651	0.605	209	11233	867	212	0.496	0.928	0.279
STD	27266	0.934	0.799	0.722	917	10324	842	438	0.677	0.925	0.589
TAb	49719	0.960	0.908	0.828	2953	7700	1558	310	0.905	0.832	0.760
PT	1878	0.956	0.427	0.395	49	12339	106	27	0.645	0.991	0.424
EU	625	0.949	0.289	0.246	38	12263	131	89	0.299	0.989	0.257
BS	305	0.994	0.311	0.325	147	12037	251	86	0.631	0.980	0.466
ST	8086	0.990	0.896	0.790	325	11851	262	83	0.797	0.978	0.653
SB	18247	0.981	0.856	0.814	959	10530	853	179	0.843	0.925	0.650
SSB	405	0.990	0.517	0.523	86	12221	149	65	0.570	0.988	0.446
SA	16394	0.952	0.747	0.681	563	9591	2188	179	0.759	0.814	0.322
PAC	10042	0.985	0.896	0.840	1769	9815	681	256	0.874	0.935	0.791
AB	273	0.997	0.800	0.593	149	11573	746	53	0.738	0.939	0.272
APNT	946	0.967	0.420	0.446	152	11701	553	115	0.569	0.955	0.313
APA	1959	0.995	0.820	0.744	249	11855	333	84	0.748	0.973	0.544
PAP	936	0.987	0.553	0.419	329	10723	1420	49	0.870	0.883	0.309
AEB	1126	0.987	0.422	0.388	647	11107	434	333	0.660	0.962	0.628
AAER	623	0.995	0.646	0.508	440	11404	579	98	0.818	0.952	0.565
AF	8463	0.988	0.878	0.787	690	11368	379	84	0.891	0.968	0.749
RAF	1941	0.994	0.699	0.545	178	12093	230	20	0.899	0.981	0.587
AFIVBC	1103	0.991	0.679	0.652	68	12343	89	21	0.764	0.993	0.553

AFL	3961	0.996	0.977	0.928	30	12255	211	25	0.545	0.983	0.203
JE	873	0.985	0.712	0.617	518	10946	819	238	0.685	0.930	0.495
JR	360	0.991	0.494	0.352	140	11759	486	136	0.507	0.960	0.310
SVT	3225	0.989	0.789	0.675	205	11562	692	62	0.768	0.944	0.352
AT	2197	0.986	0.702	0.700	25	11733	753	10	0.714	0.940	0.062
PVC	8240	0.977	0.905	0.836	1490	10392	573	66	0.958	0.948	0.823
PVPB	172	1.000	0.830	0.232	78	11781	645	17	0.821	0.948	0.191
VB	221	0.995	0.376	0.282	224	11770	508	19	0.922	0.959	0.459
VA	434	0.892	0.178	0.102	215	10957	998	351	0.380	0.917	0.242
VFB	402	0.882	0.167	0.244	45	12024	357	95	0.321	0.971	0.166
VEB	128	0.982	0.350	0.142	86	11070	1349	16	0.843	0.891	0.112
VT	91	0.984	0.506	0.233	48	12239	223	11	0.814	0.982	0.291
PES	593	0.986	0.824	0.667	110	12199	162	50	0.688	0.987	0.509
LLLV	841	0.995	0.555	0.510	83	12181	210	47	0.638	0.983	0.392
LCLV	1474	0.990	0.598	0.547	22	12336	145	18	0.550	0.988	0.213
ER	2925	0.957	0.454	0.449	34	12204	236	47	0.420	0.981	0.194
IAVB	11633	0.973	0.747	0.715	418	11414	499	190	0.688	0.958	0.548
IIAVB1	321	0.996	0.712	0.600	73	12012	416	20	0.785	0.967	0.251
IIIAVB	177	0.998	0.781	0.539	83	11652	753	33	0.716	0.939	0.174
BBB	26093	0.965	0.891	0.849	1610	9485	1108	318	0.835	0.895	0.693
LAFB	1319	0.994	0.539	0.571	515	11531	425	50	0.912	0.964	0.684
CRBBB	4844	0.985	0.831	0.729	629	11612	225	55	0.920	0.981	0.818
IRBBB	11294	0.961	0.744	0.745	358	11509	529	125	0.741	0.956	0.523
CLBBB	1224	1.000	0.955	0.839	85	12366	45	25	0.773	0.996	0.708
PRBBB	7388	0.910	0.442	0.469	17	11830	654	20	0.459	0.948	0.048
NICD	332	0.979	0.281	0.281	118	11871	499	33	0.781	0.960	0.307
IISAB	266	0.995	0.217	0.192	63	11952	486	20	0.759	0.961	0.199
HEH	16865	0.935	0.728	0.671	777	10908	538	298	0.723	0.953	0.650
HLVV	10231	0.975	0.829	0.752	727	11279	432	83	0.898	0.963	0.738
HRVV	421	0.995	0.430	0.444	87	11961	459	14	0.861	0.963	0.269
PR	2170	0.994	0.916	0.901	561	11874	55	31	0.948	0.995	0.929
APR	435	0.982	0.769	0.783	26	12465	23	7	0.788	0.998	0.634
VPR	572	0.998	0.693	0.681	267	12076	159	19	0.934	0.987	0.750
AVPR	475	1.000	0.971	0.898	85	12251	182	3	0.966	0.985	0.479
ASVPR	386	0.991	0.155	0.103	136	12211	171	3	0.978	0.986	0.610
Average	8095.5	0.975	0.646	0.575	551.4	11327.2	523.9	118.6	0.736	0.954	0.468

Table 1: Diagnosis performance of our model for 60 rhythm classes using the offline test set and online test set.

3. Please consider depositing the test set used in this study as a public dataset, for reproducibility purposes.

Response: We used an experiment to explore exactly how much ECGs should be made publicly that would reasonably test the diagnostic performance. We trained a model using the full training set ECGs and tested on 5,000, 7,000, 9,000, 11,000 and 13,000 randomly selected ECGs from the 14,418 test set data, showing that any size of test set in this range was feasible and that there was no significant difference in model performance on various size test sets. Therefore, we downscaled our offline test set to 7000 and made it publicly available.

Samples	5000	7000	9000	11000	13000	14418
AUPRC	0.650	0.646	0.649	0.647	0.644	0.645

Action: We have made the offline test set total of 7000 ECGs publicly available at <https://www.scidb.cn/s/EzMjM3>.

Reviewer #2 (Remarks to the Author):

I have no further comments. While the authors have addressed my comments appropriately, there are some fundamental shortcomings which they cannot correct. The biggest one is its generalizability in a broader population with different skin tones and muscle mass etc.

Response: ECGs from population with different skin tones and muscle mass etc. do differ significantly in certain aspects, such as physiological waveforms like the ST segment. Our data are all collected from China and the most appropriate application is within China, and it is a limitation of our model that it cannot be applied worldwide. The fundamental solution to bridge this domain gap is to collect more data, including various ethnic groups and muscle mass populations. Secondly, we need to investigate domain adaptation methods to extend the range of applications of our model.

Action: We discuss the limitations about the application of our model due to insufficient ECG sources and the method to solve the problem in the Discussion of the manuscript on page 12:

Although deep learning has been successfully applied to physiological signal processing, our model can be initially applied to clinical diagnostic settings. However, there is significant variation in ECGs due to collected from different devices, geographic areas, skin tones and muscle mass etc., our model is currently only applicable to the population using the specific wearable ECG devices within China. In addition, the number of classes in our database close to 300, many of which only have few samples such as ventricular tachycardia (VT) whose sample size is less than 100. Currently, the lack of samples is the main factor limiting the classification performance of the model to rare classes. In future work, we will expand our data sources to collect ECGs from different ethnic groups, populations and devices, and try to bridge the domain gap between the distribution of ECGs from different sources, try our best to recognize more rhythm classes at a fine-grained level and extend this work to a wider range of applications. In conclusion, we have established a baseline as far as the practicability of deep learning approach, which can provide a reference and feasible report for other researchers. The difficulties encountered by the model in real-world settings are more complex and there is much more work we can study, which provides more opportunities and challenges for the analysis and processing of ECG.

Reviewer #4 (Remarks to the Author):

Thank you for the comprehensive review of the article considering the reviewers' comments. About my suggestions, I would like to stress the following:

4. They added Supplementary Table 1 with a hierarchical list of ECG labels that answer some of the inconsistencies of the original classification.

***** However, the authors maintain using the wrong term "cardiac rhythms" for all ECG diagnostic classes. It should be clear that only when the ECG abnormality reflects an arrhythmic disorder someone can say that this is a "cardiac rhythm" class. Many of the "labels" used by the authors are standard morphological ECG abnormalities (Q wave abnormalities, high LV voltage, ST elevation). So, it is completely wrong to say that these abnormalities are "cardiac Rhythm" classes. This must be changed in a final version.**

***** There are inconsistencies and unacceptable absences in the list of the Supplementary Table 1. So there is no complete AV block, and the third column does not match very well with the 2nd one.**

Response: We value your feedback and comments. We researched some literature and decided to follow the article *Cardiologist-level arrhythmia detection and classification in ambulatory electrocardiograms using a deep neural network* by Hannun A Y et al. in Nature medicine and use "rhythm classes" as the generic term. In addition, we have amended Supplementary Table 1 in accordance with your suggestions.

Action: In the manuscript we use the term "rhythm classes" as the generic term for all ECG diagnostic classes, and we have listed all the ECG labels in Supplementary Table 1 and amended it in response to your comments of the Supplementary Information.

5. I wrote: "The general accuracy of the algorithm cannot be calculated as the average of the accuracy for each ECG diagnosis. As the authors pointed out, each ECG can have more than one ECG abnormality and the precision should be evaluated for the ECG tracing itself." The authors did not provide the general accuracy of the method considering the right diagnosis per ECG, not per ECG class. Please, provide it.

Response: Our previous metrics for evaluating the diagnostic performance of the model used mainly label-based metrics. We have taken your suggestion and added 4 example-based metrics for the objective case of multiple labels on one single ECG: Sen_{exam} , Spe_{exam} , $F1_{exam}$ and Acc_{exam} .

Action: We have added the analysis and Table 2 on page 7 of the manuscript:

Intelligent diagnosis of ECGs in real clinical setting is a multi-label classification task, where a cardiologist may annotate multiple labels to a single ECG. Therefore, we evaluate the classification performance of the model in example wise. Our model was able to detect and classify 60 labels. In the large-scale online test, each ECG had an average of 3.3 labels, and the intelligent diagnostic model was able to detect 2.7 of them, missing 0.6, while bringing in 2.5 false positives that were acceptable. This demonstrates that our model is able to diagnose most of the labels correctly, transforming the diagnostic task of cardiologists from a fill-in-the-blank to a multiple-choice and review, and can effectively reduce their workload.

Intelligent	TP_{exam}	TN_{exam}	FP_{exam}	FN_{exam}	Sen_{exam}	Spe_{exam}	$F1_{exam}$	Acc_{exam}
Offline	3.0	56.1	1.4	0.5	0.873	0.975	0.756	0.969
Online	2.7	55.3	2.5	0.6	0.857	0.956	0.653	0.950

Table 2: Diagnosis performance of our model in example wise using the offline test set and online test set.

REVIEWERS' COMMENTS

Reviewer #4 (Remarks to the Author):

All modifications made by the authors have significantly improved the study. However, there is still one point that should be improved. "Rhythm classes" refer to specific ECG diagnoses related to cardiac arrhythmias. Hannun A Y et al. in Nature Medicine used "rhythm classes" as the generic term because their study was on arrhythmia detection and classification. In other words, they studied only cardiac arrhythmias in ambulatory electrocardiograms. The authors of the present study are studying not only cardiac arrhythmias but also other ECG abnormalities that are not related to cardiac arrhythmias, such as bundle branch blocks, LV hypertrophy and ST elevation. These abnormalities are not arrhythmias and cannot be referred as "rhythm classes". So, it is wrong to say that all these abnormalities are "rhythm classes". As a suggestion, refer to standard guidelines on reporting to use the appropriate nomenclature:

Mason JW, Hancock EW, Gettes LS, Bailey JJ, Childers R, Deal BJ, Josephson M, Kligfield P, Kors JA, Macfarlane P, Pahlm O, Mirvis DM, Okin P, Rautaharju P, Surawicz B, van Herpen G, Wagner GS, Wellens H; American Heart Association Electrocardiography and Arrhythmias Committee, Council on Clinical Cardiology; American College of Cardiology Foundation; Heart Rhythm Society. Recommendations for the standardization and interpretation of the electrocardiogram: part II: electrocardiography diagnostic statement list a scientific statement from the American Heart Association Electrocardiography and Arrhythmias Committee, Council on Clinical Cardiology; the American College of Cardiology Foundation; and the Heart Rhythm Society Endorsed by the International Society for Computerized Electrocardiology. J Am Coll Cardiol. 2007 Mar 13;49(10):1128-35. doi: 10.1016/j.jacc.2007.01.025. PMID: 17349897.

REVIEWER COMMENTS

Reviewer #4 (Remarks to the Author):

All modifications made by the authors have significantly improved the study. However, there is still one point that should be improved. "Rhythm classes" refer to specific ECG diagnoses related to cardiac arrhythmias. Hannun A Y et al. in *Nature Medicine* used "rhythm classes" as the generic term because their study was on arrhythmia detection and classification. In other words, they studied only cardiac arrhythmias in ambulatory electrocardiograms. The authors of the present study are studying not only cardiac arrhythmias but also other ECG abnormalities that are not related to cardiac arrhythmias, such as bundle branch blocks, LV hypertrophy and ST elevation. These abnormalities are not arrhythmias and cannot be referred as "rhythm classes". So, it is wrong to say that all these abnormalities are "rhythm classes". As a suggestion, refer to standard guidelines on reporting to use the appropriate nomenclature:

Mason JW, Hancock EW, Gettes LS, Bailey JJ, Childers R, Deal BJ, Josephson M, Kligfield P, Kors JA, Macfarlane P, Pahlm O, Mirvis DM, Okin P, Rautaharju P, Surawicz B, van Herpen G, Wagner GS, Wellens H; American Heart Association Electrocardiography and Arrhythmias Committee, Council on Clinical Cardiology; American College of Cardiology Foundation; Heart Rhythm Society. Recommendations for the standardization and interpretation of the electrocardiogram: part II: electrocardiography diagnostic statement list a scientific statement from the American Heart Association Electrocardiography and Arrhythmias Committee, Council on Clinical Cardiology; the American College of Cardiology Foundation; and the Heart Rhythm Society Endorsed by the International Society for Computerized Electrocardiology. *J Am Coll Cardiol.* 2007 Mar 13;49(10):1128-35. doi: 10.1016/j.jacc.2007.01.025. PMID: 17349897.

Response: Thank you for your careful review and the recommendation of professional references. It is true that the diagnosable classes of our model include overall interpretation such as normal ECG, various waveform changes, various arrhythmias, axis, voltage and heart enlargement and hypertrophy, which is inappropriate use the "rhythm classes" as a generic terminology. Therefore, we have carefully studied the literature and decided to replace the generic terminology for all classes as "diagnostic terms for ECG".

Action: We have replaced the "arrhythmia detection algorithm" in the title with "intelligent diagnostic algorithm", and have replaced all "rhythm classes" in the main text with "diagnostic terms for ECG".